# All-optical phase conjugation using diffractive wavefront processing

Che-Yung Shen [1,2,3,4], Jingxi Li [1,2,3,4], Tianyi Gan [1,3], Yuhang Li[1,2,3], Mona Jarrahi [1,3] & Aydogan Ozcan [1,2,3] ✉

Optical phase conjugation (OPC) is a nonlinear technique used for counteracting wavefront distortions, with applications ranging from imaging to beam focusing. Here, we present a diffractive wavefront processor to approximate all-optical phase conjugation. Leveraging deep learning, a set of diffractive layers was optimized to all-optically process an arbitrary phase-aberrated input field, producing an output field with a phase distribution that is the conjugate of the input wave. We experimentally validated this wavefront processor by 3D-fabricating diffractive layers and performing OPC on phase distortions never seen during training. Employing terahertz radiation, our diffractive processor successfully performed OPC through a shallow volume that axially spans tens of wavelengths. We also created a diffractive phase-conjugate mirror by combining deep learning-optimized diffractive layers with a standard mirror. Given its compact, passive and multi-wavelength nature, this diffractive wavefront processor can be used for various applications, e.g., turbidity suppression and aberration correction across different spectral bands.

Optical phase conjugation (OPC) has been used to counteract wavefront distortions by synthesizing a wave that is the complex conjugate of an impinging distorted wave, which can retrace the original propagation path and thereby "undo" the effects of wavefront-induced distortions[1]. Over half a century ago, a seminal work unveiled that OPC could counteract optical scattering using a photorefractive crystal as the phase-conjugate mirror[2]. Phase conjugation has also been demonstrated by degenerate four-wave mixing in absorbing materials[3,4]. This unique technique later fostered myriad applications including, but not limited to, laser beam focusing through scattering media[5–16], imaging through turbid materials[17–20], compensation of aberrations in microscopy[21], and improving the performance of optical communication systems[22–26], among others. As one implementation of OPC, analog optical phase conjugation (AOPC)[5,6,9,15,17] harnesses nonlinear materials such as photorefractive materials to store the scattered field and produce its phase conjugate, presenting the merits of rapid and continuous

phase modulation. Nevertheless, the predominant drawback of AOPC stems from its low conjugation reflectivity, leading to relatively weak energy in the time-reversed beam. As an alternative, digital optical phase conjugation (DOPC) methods[8,10,12–14,16,18,19,27] offer a different approach that uses a digital camera in an interferometric set-up to capture the optical wavefront information and digitally produces an OPC field, which can provide much higher conjugation reflectivity. However, DOPC is challenged with its intrinsically slower response, which results in playback latencies of tens of milliseconds and discrete modulation units on the order of several micrometers. In addition, the requirement of high-sensitivity interferometric/holographic imaging set-ups and light field modulation components such as spatial light modulators (SLMs) also bring considerable cost, system complexity, and large footprint to existing DOPC solutions. Moreover, the digital recovery and processing of the optical phase information also add to the computational load of these DOPC systems.

[1]Electrical and Computer Engineering Department, University of California, Los Angeles, CA, USA. [2]Bioengineering Department, University of California, Los Angeles, CA, USA. [3]California NanoSystems Institute (CNSI), University of California, Los Angeles, CA, USA. [4]These authors contributed equally: Che-Yung Shen, Jingxi Li. ✉e-mail: ozcan@ucla.edu

Deep learning-based engineering of materials has recently emerged as a new approach for the inverse design of optical systems for various non-intuitive and nontrivial functions[28]. By leveraging multiple cascaded diffractive layers that are spatially engineered with wavelength-scale features, optimized using deep learning, a diffractive optical network (or a diffractive deep neural network, D²NN)[29–42] can be formed to perform universal linear transformations through the modulation of optical fields, and has been demonstrated for various applications including multispectral imaging[43], quantitative phase imaging[35], unidirectional imaging[39], pulse shaping[44], spectral filtering[45], etc. The unique advantage of this diffraction-based visual computing platform lies in its ability to allow the thin diffractive material to directly interact with the input field wavefront and process the light field in situ, thereby significantly reducing or even eliminating the need for digital post-processing. Also, the processing is performed all optically, using passive materials that do not consume power.

Here, we introduce an all-optical phase conjugation framework based on deep learning-engineered diffractive optical structures, which collectively perform phase conjugation on optical fields with unknown phase aberrations. The OPC task is completed at the speed of light propagation through a thin diffractive volume that axially spans ~84 λ, where λ denotes the operating wavelength of the system. This OPC framework comprises multiple successively positioned dielectric diffractive layers, where each layer is spatially coded with hundreds of thousands of diffractive features, each engineered at a half-wavelength lateral pitch. Once optimized through deep learning in a computer, these diffractive layers are 3D fabricated to form a physical wavefront processor capable of processing a phase-distorted input wavefront that passes through its input aperture. This diffractive all-optical processing results in a phase profile at the output aperture, representing a conjugated version of the input phase distribution. Due to the use of passive optical modulation components in this diffractive processor, the OPC operation is solely driven by the intrinsic power of the input optical field, eliminating the need for any digital processing or other external power sources except for the illumination light. Moreover, since our diffractive OPC system accomplishes its task in an end-to-end manner as the field propagates through a very thin diffractive volume, it obviates the processing speed constraints commonly imposed by digital computation and light field modulation set-ups employed in traditional DOPC solutions, thus enabling ultra-high-speed phase conjugation operations at nanosecond or even picosecond levels, depending on the operation wavelength.

To demonstrate the efficacy of our framework, we first designed a transmissive diffractive OPC processor composed of 8 diffractive layers to perform phase conjugation at a single wavelength within a short axial length of ~84 λ. To train this diffractive processor design, we created a training dataset by randomly generating phase distributions composed of Zernike polynomials as the input field profiles, each paired with its phase-conjugated counterpart as the ground truth (desired profile) of the output phase. After the training, we conducted numerical blind testing using unseen input fields; when the trained diffractive processor was tested with phase aberrated input fields composed of 2 randomly selected Zernike polynomials, its output fields had a low phase mean absolute error (MAE) of <1.5%. Moreover, when testing using input phase profiles composed of 3 randomly selected Zernike polynomials, representing different types of phase distortions not included in the training inputs, the resulting phase MAE values still remained <1.5%, highlighting the diffractive processor's adeptness in processing unknown, complicated phase aberrated input wavefronts. Our numerical analyses revealed that, for a range of input phase contrast, our diffractive processor can be regarded as an approximation of the desired OPC function with minimal errors; however, as the input phase contrast and level of phase aberrations escalate, this approximation becomes less accurate. Furthermore, we numerically demonstrated the "time-reversal" effect of this diffractive

OPC framework, which was achieved by symmetrically placing the same phase distortion plane before and after the transmissive diffractive OPC design. In our blind testing, the diffractive OPC processor successfully showed phase conjugation operation on a plane wave disturbed by an unknown, random phase perturbation, proving the utility of our diffractive OPC processor. In addition, we demonstrated our diffractive processor's capability for multi-wavelength OPC operation, showcasing its practicality for applications spanning different spectral bands.

As an experimental proof-of-concept demonstration, we validated our diffractive OPC framework at the terahertz (THz) part of the spectrum. Leveraging 3D printing, we fabricated a 3-layer diffractive OPC processor, trained to perform OPC operation on phase aberrated input wavefronts at a wavelength of 0.75 mm. This fabricated diffractive processor was positioned centrally between two identical unknown phase perturbation planes that were also 3D-printed and then tested with an input pinhole object to perform all-optical phase conjugation of aberrated spherical wavefronts. Although the diffractive OPC processor never saw the testing phase aberrations during its training, our experiments culminated in the successful reconstructions of sharply focused spots at the output, mirroring the original image of the pinhole object, experimentally verifying the success of our all-optical phase conjugation operation. Beyond this single-wavelength diffractive OPC design, we also trained and fabricated a diffractive multi-wavelength OPC design and demonstrated its successful operation at three distinct wavelengths (0.75, 0.775, and 0.8 mm), experimentally validating the feasibility of our broadband OPC framework in generating multi-wavelength phase-conjugated fields.

Delving deeper, we also explored combining the diffractive OPC processor with a reflective mirror to form a diffractive phase conjugate mirror. By placing a flat mirror behind the diffractive processor volume, the output wavefront from the diffractive processor folds back into the same diffractive layers (i.e., a double-pass configuration), ultimately reaching the same aperture used at the input. Our numerical results indicate that this diffractive phase conjugate mirror can be successfully trained to perform phase conjugation of unknown, randomly distorted wavefronts, with error levels almost identical to the transmissive OPC configuration.

In this work, we demonstrate a diffractive processor to all-optically perform phase conjugation operation on input optical fields. This diffractive OPC platform holds design flexibility since it can function at different parts of the electromagnetic spectrum without retraining or optimizing its diffractive layers by simply scaling its features in proportion to the illumination wavelength. This feature makes our diffractive OPC framework highly desirable for correcting wavefront distortions at various parts of the spectrum where cost-effective and easy-to-implement phase conjugation solutions do not readily exist, such as the IR and THz bands. Moreover, since isotropic dielectric materials are used to fabricate these diffractive layers, the OPC function is independent of the polarization state of the incident light, which remains unchanged at the output of the diffractive OPC system. Combining these unique advantages, our diffractive OPC framework holds promise for various applications, including e.g., turbidity suppression and phase aberration correction, and can unlock different opportunities across diverse areas, including but not limited to biomedical imaging, microscopy, telescope systems, and optical communication.

## Results
### Design of a transmissive diffractive OPC processor
Figure 1 illustrates the general concept of a diffractive OPC processor that operates in transmission, i.e., with an input and an output aperture positioned at the different sides of the diffractive processor. As depicted in Fig. 1a, a diffractive OPC network composed of *K*

successive diffractive layers ($L_1, ..., L_K$) is positioned between the input and output apertures, with its primary function to perform optical phase conjugation of the aberrated fields within the input aperture and relay the resulting phase-conjugated fields into the output aperture. Each of these diffractive layers is coded with the same number of spatially engineered features, each with a lateral width of ~$\lambda/2$ and a trainable thickness that provides a full phase modulation covering 0 to $2\pi$. The input aperture, diffractive layers, and output aperture are connected to each other through free space. The intended functionality of this diffractive OPC framework is further elaborated on in Fig. 1b. An incident phase aberrated field $i = e^{j\Psi}$ with a uniform amplitude and an unknown phase profile $\Psi$ passes through the input aperture. After being processed by the diffractive OPC processor, the resulting complex field is collected by the output aperture, represented as $o = \mathrm{D^2NN}\{i = e^{j\Psi}\}$. The objective function of our diffractive OPC processor is to ensure that, for any phase aberrated input field $i = e^{j\Psi}$, the diffractive output field $o$ would exhibit a phase profile that closely approximates the phase-conjugated version of the input field, i.e., $o \approx \alpha \cdot o^{(\mathrm{GT})} = \alpha \cdot i^* = \alpha \cdot e^{-j\Psi}$, where GT stands for ground truth, and

$\alpha$ is a scalar constant that accounts for the output diffraction efficiency of the OPC processor.

It is worth noting that, in the analyses presented in this paper, we demonstrated OPC operations with a unit magnification between the input and output apertures, axially separated by several tens of wavelengths. This design choice ensures a relatively high diffraction efficiency within the output aperture of the OPC processor and also allows the precise modeling of the free space propagation between adjacent diffractive planes using the angular spectrum method[46], with a lateral sampling of ~$\lambda/2$ at the input and the propagated fields. If phase conjugation with magnification or demagnification at the output field is desired, the implementation of a pyramidal architecture for the diffractive OPC processor[47] may be considered. In that case, the free-space propagation can be modeled using a scalable angular spectrum method[48] or the Fresnel diffraction approach to enable different sampling sizes at the input and output fields. These approaches can allow for the efficient design and modeling of diffractive OPC processors, accommodating a wider range of structural parameters and applications.

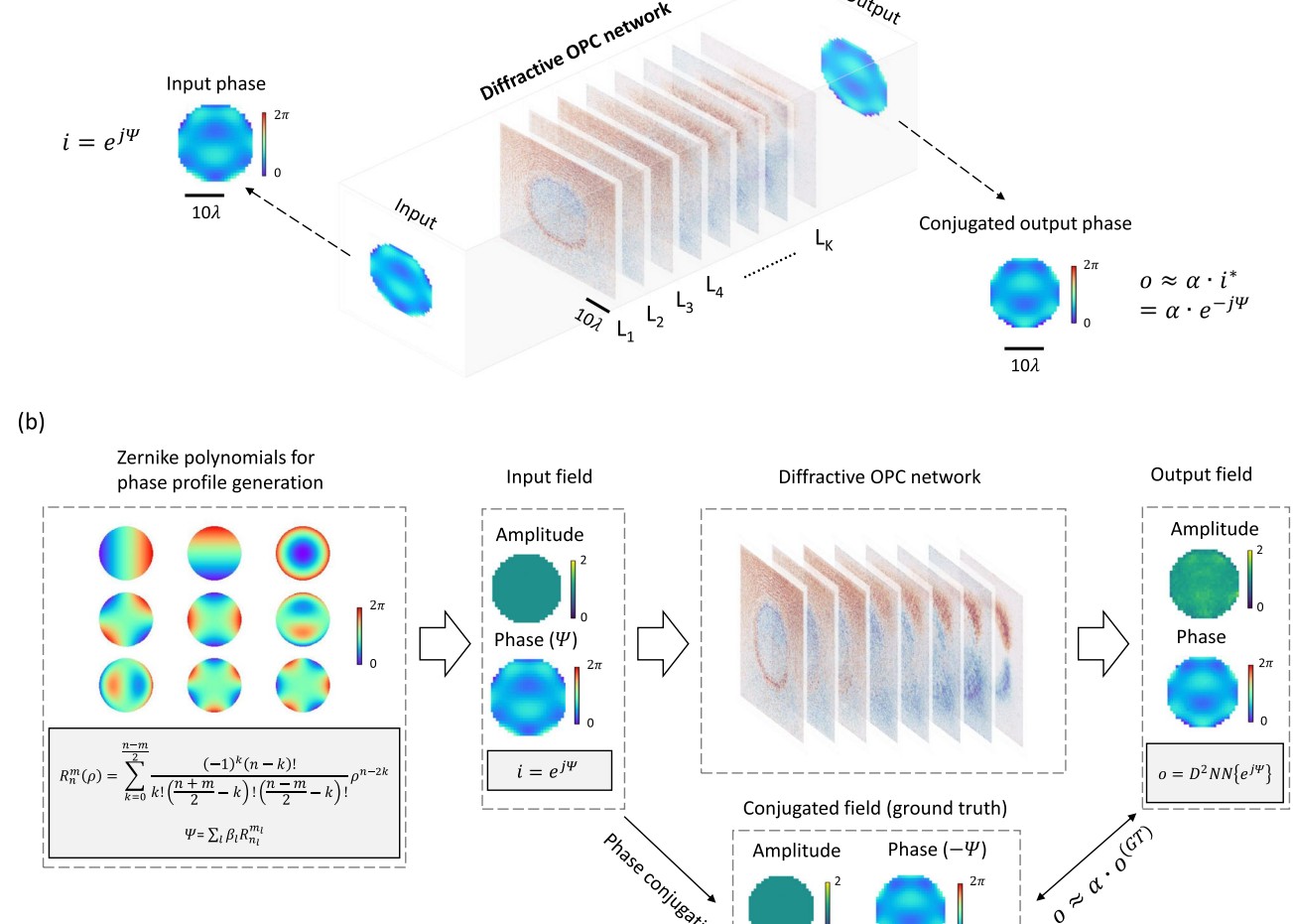

**Fig. 1 | Schematic and operation mechanism of a transmissive diffractive optical phase conjugation (OPC) processor. a** Optical layout of a diffractive OPC processor that operates in transmission geometry. This OPC processor is composed of $K$ passive diffractive layers ($L_1$–$L_K$), jointly trained using deep learning to perform OPC of an incoming complex field $i = e^{j\Psi}$ with unknown phase aberrations $\Psi$. **b** Pipeline of the demonstrated diffractive OPC framework. The resulting output complex field $o$ within the output aperture represents the phase-conjugated version of the input field, i.e., $o \approx \alpha \cdot o^{(\mathrm{GT})} = \alpha \cdot i^* = \alpha \cdot e^{-j\Psi}$.

As a proof-of-concept demonstration, we initially chose to conduct numerical simulations of the diffractive OPC processor within the terahertz part of the spectrum, at a wavelength of $\lambda = 0.75$ mm. Furthermore, we leveraged phase distributions formed by Zernike radial polynomials as our testbed to assess the phase conjugation capability of our diffractive OPC framework. Zernike radial polynomials are a set of continuous functions orthogonal over a unit circle[49], which can be used to describe typical wavefront aberrations or deviations from an ideal wavefront in various optical systems. The mathematical representation of a Zernike radial polynomial is given by:

$$R_n^m(\rho) = \sum_{k=0}^{\frac{n-m}{2}} \frac{(-1)^k (n-k)!}{k! \left(\frac{n+m}{2} - k\right)! \left(\frac{n-m}{2} - k\right)!} \rho^{n-2k} \quad (1)$$

Here, $\rho$ denotes the normalized radial coordinate, constrained within the range of [0, 1]. Both $m$ and $n$ are nonnegative integers, collectively defining a specific mode of the Zernike polynomial. Practically, the polynomials $\{R_n^m\}$ serve as a basis set for characterizing wavefront aberrations, with each $R_n^m$ corresponding to a distinct type of aberration. For our study, we utilized randomly generated combinations of these Zernike polynomials, i.e., $\sum_l \beta_l R_{n_l}^{m_l}$, to create a variety of phase distributions. These phase distributions were used as the phase profiles of the input fields $i = e^{j\psi}$ within a circular input aperture, to be processed by our diffractive OPC framework.

To make our diffractive phase conjugation framework successful, it is imperative to train our diffractive model to accommodate a wide variety of phase profiles. With this in mind, we generated a set of 200,000 randomly selected Zernike polynomial phase distributions for training proposes. Each of these phase profiles was formulated by randomly selecting two from the first 28 Zernike polynomials and linearly combining them using random weight coefficients, which can be described as:

$$\Psi = \beta_1 R_{n_1}^{m_1} + \beta_2 R_{n_2}^{m_2} \quad (2)$$

where $\beta_1 \neq 0$ and $\beta_2 \neq 0$ are the weight coefficients of the polynomials adhering to the constraint that $\beta_1 + \beta_2 = \alpha_{tr}\pi$. During the training data generation, the coefficient $\beta_1$ for each $\Psi$ was randomly chosen from the range $[0.1\alpha_{tr}\pi, 0.9\alpha_{tr}\pi]$. The polynomial mode numbers, $(m_1, m_2)$ and $(n_1, n_2)$, were also randomly selected within the ranges of [−6, 6] and [0, 6], respectively, ensuring the selection only from the first 28 Zernike polynomials. With this formulation, each $\Psi$ possesses a dynamic range of $[0, \alpha_{tr}\pi]$, where $\alpha_{tr}$ denotes a training phase contrast parameter. To facilitate the diffractive OPC network's capability to perform all-optical phase conjugation across different input phase contrast values, $\alpha_{tr}$ was randomly chosen between 0.2 and 1, i.e., $\alpha_{tr} \in U[0.2, 1]$. Based on this diverse set of $\Psi$, we also generated their conjugated versions as the training target (ground truth) of our diffractive OPC processor, thus forming a training dataset consisting of 200,000 input/target complex field pairs.

We trained our diffractive OPC network with $K = 8$ successive diffractive layers using an operational wavelength of $\lambda = 0.75$ mm, as illustrated in Fig. 2a. During the training process, the thickness profiles of the diffractive layers were iteratively optimized via error-backpropagation and stochastic gradient descent techniques (see the "Methods" section for details). In this work, the necessity for precise control over the output complex field presents a significant challenge in diffractive optical information processing, approximating a nonlinear OPC function that has not been explored yet. To address this challenge, we created a new optimization process to minimize a specially devised loss function (detailed in Methods), which compares both the normalized amplitude and phase differences between the diffractive output fields $o$ and their corresponding ground truth $o^{(GT)}$. More details regarding the training loss function, numerical forward

model and structural parameters of the diffractive OPC processor can be found in the Methods section.

## Performance analysis of a transmissive diffractive OPC processor

After the deep learning-based optimization of the diffractive OPC processor, the resulting diffractive layer thickness profiles are visualized in Fig. 2b. To blindly test the OPC performance of our trained design, we first created a test set containing 10,000 pairs of input/target fields by following the same approach that we used for constructing the training set, except that each input/target phase profile in this test set has a phase contrast of $\alpha_{test} = 1$. Stated differently, the input fields in this test set can be represented as $i = e^{j\psi}$ and $\Psi = \beta_1 R_{n_1}^{m_1} + \beta_2 R_{n_2}^{m_2}$, where $\beta_1 + \beta_2 = \pi$, $\beta_1 \neq 0$ and $\beta_2 \neq 0$, indicating that all the input/target fields have a dynamic phase range of $[0, \pi]$. These input fields in the test set were also randomly generated to be different from those in the training set, ensuring that they were never seen before by our trained diffractive model.

Based on these test input fields, we obtained their corresponding output fields produced by the diffractive OPC network through numerical simulations. To evaluate these results, we first normalized these output complex fields with respect to their ground truth, eliminating the effect of their scaling mismatch caused by the output diffraction efficiency. Following that, we computed the MAE values for both the phase and amplitude components of the normalized output fields compared to the phase-conjugated ground truth, which we termed phase and amplitude MAEs, respectively. Based on this evaluation approach, we achieved an average phase MAE of $1.38 \pm 0.12\%$ across the entire test set. This suggests that the phase profiles of the output fields generated by our diffractive OPC network align closely with the target phase conjugated output distribution, manifesting only minimal discrepancies. On the other hand, the average amplitude MAE was quantified as $8.89 \pm 1.91\%$, which also represents a relatively low error level, despite being larger compared to its phase counterpart. Figure 2c provides examples of diffractive output fields alongside their respective ground truth and error maps. These results illustrate that the diffractive output fields exhibit phase distributions almost identical to their ground truth. The amplitude of these output fields also presents a good uniformity across the output aperture of the diffractive OPC design.

In addition to these blind testing results, we created another test set of phase aberrated input fields with their phase profiles constituted by only a single, randomly selected Zernike polynomial term, i.e., $\Psi = R_n^m$. Such phase distributions, which represent a particular case of Eq. (2) with $\beta_1$ or $\beta_2 = 0$, can be used to describe distinct types of phase aberrations, including those frequently occurring in the pupil function of imaging systems such as *coma* and *astigmatism*[50]. We blindly tested our diffractive OPC network using 28 of such input fields (randomly generated), where each field corresponds to one of the first 28 Zernike polynomial terms - never seen by the diffractive model during the training process. The resulting diffractive output fields based on these input fields were then quantitatively compared with their phase-conjugated target fields, which yielded phase and amplitude MAE values of $1.71 \pm 0.14\%$ and $13.13 \pm 2.31\%$, respectively, revealing a decent blind testing performance. This success is also confirmed by visualizing some diffractive output fields, as shown in Fig. 2d, which correspond to $R_2^{-2}$, $R_3^{-1}$, $R_4^0$ and $R_3^{-3}$, representing coma, astigmatism, spherical aberration and trefoil, respectively[51]. Here, the output fields of the diffractive OPC processor present a successful phase conjugation operation performed on these typical phase aberrations, despite some imperfections in their amplitude components in regions with larger phase values, echoing our prior observations in Fig. 2c.

Next, we blindly tested our diffractive OPC model using input fields with their phase profiles constituted by a combination of 3 randomly selected Zernike polynomials, i.e., $\Psi = \beta_1 R_{n_1}^{m_1} + \beta_2 R_{n_2}^{m_2} + \beta_3 R_{n_3}^{m_3}$

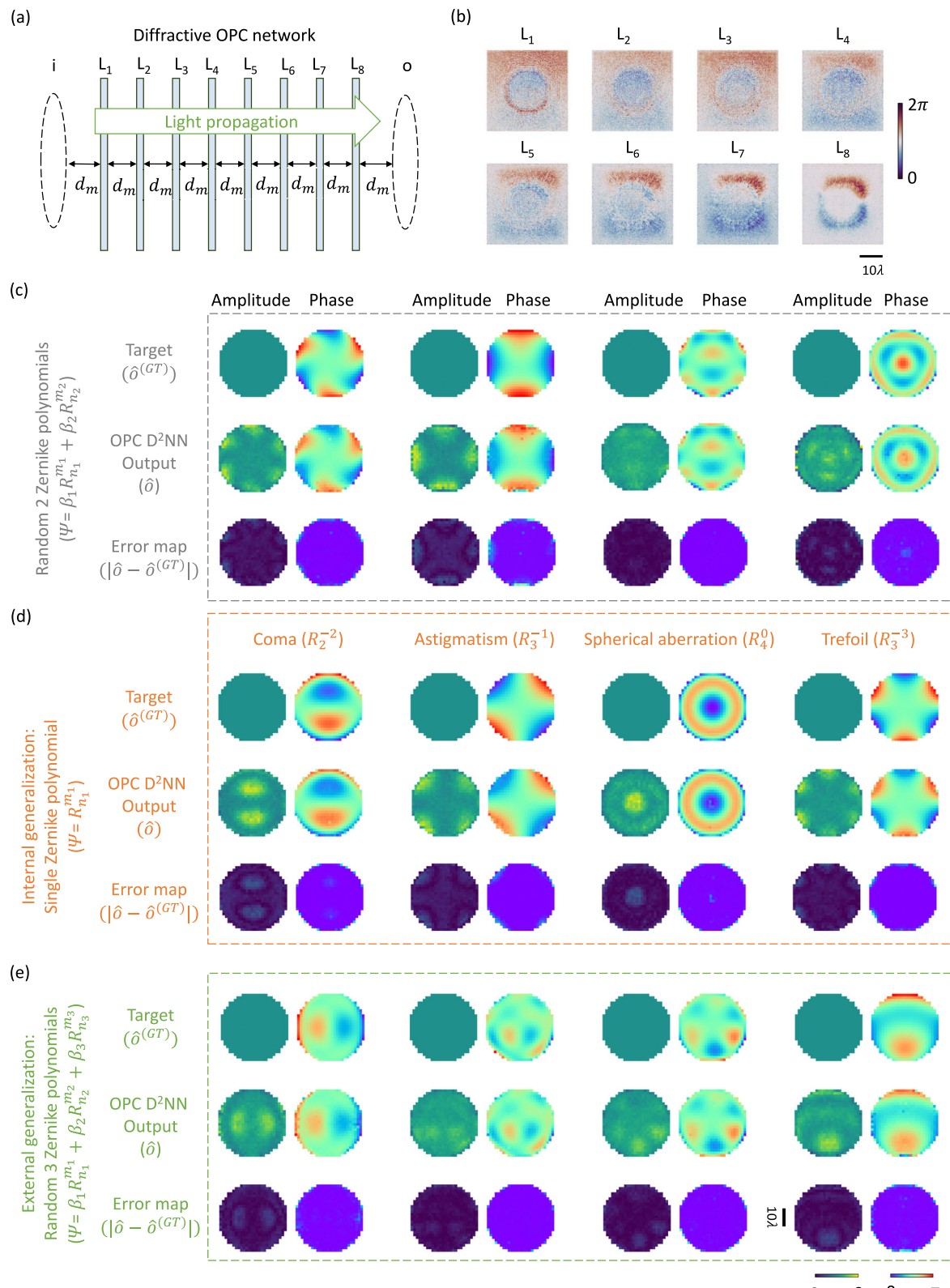

**Fig. 2 | The transmissive diffractive OPC processor design and the visualization of the diffractive output field examples. a** Illustration of a transmissive diffractive OPC processor. **b** Thickness profiles of the resulting diffractive layers trained through deep learning. **c** Amplitude and phase profiles of the exemplary output complex fields produced by the diffractive OPC processor subject to phase aberrated input fields. These input fields possess phase profiles constituted by two random Zernike polynomial terms, never seen during the training stage. For each of these diffractive output fields, its ground truth with perfect phase conjugation is also shown, along with the error map visualizing the absolute amplitude and phase differences between the output field and the ground truth. **d** Same as (**c**), but the used input fields are constituted by only one random Zernike polynomial term, never seen during the training stage. **e** Same as (**c**) and (**d**), but the used input fields are constituted by three random Zernike polynomial terms, never seen during the training stage.

(i.e., $\beta_1 + \beta_2 + \beta_3 = \pi$, where $\beta_1 \neq 0$, $\beta_2 \neq 0$ and $\beta_3 \neq 0$). This test set demonstrates even more complicated phase structures than those seen during the training. Using randomly generated 10,000 input test fields, the same diffractive OPC processor design achieved phase and amplitude MAE values of $1.09 \pm 0.06\%$ and $8.39 \pm 1.32\%$, respectively, further demonstrating its generalization success, performing all-optical phase-conjugation on various forms of randomly generated phase aberrations. The visualization of exemplary output fields is shown in Fig. 2e, underscoring our framework's robust external generalization capability for more complex phase conjugation tasks that were never represented in the training process.

In the analyses reported so far, our performance evaluation was conducted using phase-aberrated input fields with $\alpha_{\text{test}} = 1$. To delve deeper into the effects of varying $\alpha_{\text{test}}$ on the performance of OPC, we extended our analysis across an array of $\alpha_{\text{test}}$ values: [0.2, 0.4, 0.6, 0.8, 1, 1.2, 1.4, 1.6, 1.8, 1.99], using the same diffractive OPC model trained with $\alpha_{\text{tr,max}} = 1$. For this analysis, we used randomly generated input fields with phase profiles characterized by a single Zernike polynomial, i.e., $\Psi = R_n^m$, which corresponds to one of the first 28 Zernike polynomial terms, never seen by the diffractive OPC model during its training process. As shown in Fig. 3a, the resulting normalized amplitude and phase MAE values, plotted as a function of $\alpha_{\text{test}}$, reveal a rising error trend as $\alpha_{\text{test}}$ increases. Notably, when $\alpha_{\text{test}} < 1$, our diffractive model consistently delivers phase MAE values below 2%, showcasing its adeptness at phase conjugation tasks in this $\alpha_{\text{test}}$ range, which falls within our training ($\alpha_{\text{tr,max}} = 1$). For example, for $\alpha_{\text{test}} = 0.2$, the phase MAE value drops to $0.14 \pm 0.03\%$, highlighting the diffractive OPC model's excellent performance for a smaller range of phase aberrations. However, for $\alpha_{\text{test}} > 1$ (which is outside of our training range), error margins widen considerably, with the most pronounced increase observed at $\alpha_{\text{test}} = 1.99$, where the phase MAE value escalates to $6.83 \pm 0.78\%$, a value nearly four times greater than its counterpart at $\alpha_{\text{test}} = 1$, which stands at $1.71 \pm 0.14\%$. These observations are accentuated in Fig. 3b, which depicts the diffractive output fields at $\alpha_{\text{test}} = 0.2, 1.0$ and $1.8$.

For larger phase aberrations corresponding to out-of-distribution test values with $\alpha_{\text{test}} > 1$, there is a notable decline in the accuracy of the amplitude and phase output profiles of the diffractive OPC processor. This phenomenon stems from the fact that the universal linear transformation capability of a diffractive design provides a less accurate approximation for the OPC operation, which increases the output errors for larger phase aberration values corresponding to $\alpha_{\text{test}} > 1$. On the other hand, the diffractive outputs of the OPC processor almost identically mirror the desired phase-conjugated ground truth when the phase contrast is relatively small, e.g., $\alpha_{\text{test}} < 1$.

## Correction of phase aberration-induced wavefront distortions using a diffractive OPC processor

To shed more light on the capabilities of our diffractive OPC system in performing all-optical phase conjugation, we tested the time-reversal effect of OPC to counteract wavefront distortions caused by random unknown phase aberrations, as depicted in Fig. 4a. In this scenario, two-phase perturbation planes that exhibit identical but randomly selected, unknown phase aberrations were positioned, before and after a diffractive OPC system. Upon encountering the first phase aberration plane, an incoming plane wave has its wavefront distorted in an unknown, random manner. Subsequently, after passing through the diffractive OPC system, the field of the outcoming wave is conjugated, which then impinges on the second phase aberration plane. Ideally, for a perfect OPC device, the phase-conjugated wavefront, after passing through the second phase aberration plane, should manifest as a uniform plane wave – cleaned from any aberrations. To validate/test this behavior of our diffractive OPC design, we selected coma, astigmatism, spherical aberration, and trefoil as different forms of random phase aberrations and employed the same diffractive OPC

design shown in Fig. 2b, without any additional optimization. The results of this analysis are presented in Fig. 4b, which revealed a decent output phase profile in each case, cleaned from the aberrations of coma, astigmatism, spherical aberration, and trefoil. We also contrasted these findings with the results obtained without the diffractive OPC processor between the two aberration planes – this resulted in non-uniform phase aberrations at the output plane, as expected. To better comprehend these results from an application standpoint, we further focused these output fields through a diffraction-limited lens, the results of which are summarized in Fig. 4c. This aligns more closely with some of the existing practical applications where an OPC system is used to correct distorted wavefronts due to e.g., scattering media, enabling the conjugated wave to refocus through the same scattering medium again. The results reported in Fig. 4c corroborate the diffractive OPC system's efficacy in achieving precise and sharp focusing. In contrast, in the absence of a diffractive OPC processor, apparent speckles can be found around the output focal point. These analyses further illustrate the efficacy of our diffractive OPC system, highlighting its potential applications for beam focusing through aberrations.

## Optical phase conjugation of multi-wavelength illumination

Our diffractive OPC processor model previously presented in Fig. 2 was designed solely for operation at a single wavelength and was numerically demonstrated at the terahertz part of the spectrum. In the following numerical analyses, we present designs of diffractive OPC processors that are capable of multispectral operation and demonstrate their efficacy within the visible spectrum (400–750 nm), where at each wavelength channel, the aberrated input wavefronts are independent of the other wavelength channels, representing a challenging broadband OPC task. As illustrated in Supplementary Fig. S1a, these multi-wavelength diffractive designs simultaneously perform phase conjugation of aberrated input wavefronts at $N_w$ distinct wavelengths $\{\lambda_1, \lambda_2, \ldots, \lambda_{N_w}\}$, uniformly distributed within a range of 400 nm to 750 nm. In these broadband diffractive OPC processor designs, we used $N_w = 2, 4, and\ 8$. Given this wavelength multiplexing task at hand, the total number of trainable diffractive features ($N$) in the OPC processor was scaled proportionally with the number of wavelength channels ($N_w$) to maintain the information processing capability per channel. To analyze this behavior for each $N_w$ choice, we created different OPC processors with $N = \{0.5 N_i N_o N_w, N_i N_o N_w, 2 N_i N_o N_w\}$ and $K = 12$. Here, $N_i$ and $N_o$ represent the number of diffraction-limited pixels within the input and output apertures, respectively. In these numerical analyses, we selected N-BK7 glass as the diffractive material due to its prevalent use for optical components at the visible band[52]. The training data and methods used for these visible band diffractive OPC processor designs follow their terahertz counterparts. After the training, the diffractive layer thickness profiles corresponding to the model with $N = 2 N_i N_o N_w$ and $N_w = 8$ wavelengths are visualized in Supplementary Fig. S1b as an example.

To evaluate the performance of these diffractive multi-wavelength OPC processors, we performed numerical blind testing using input fields with phase profiles composed of 2 random Zernike polynomials, which randomly varied at different wavelengths and were never used in the training. The average phase errors for all these diffractive OPC models with different $N$ and $N_w$ values are summarized on the left part of Supplementary Fig. S2, revealing an error level of <1.5% for all the diffractive models. These results also indicate that introducing more degrees of freedom in the design (by increasing $N$) can substantially improve the phase accuracy of multispectral OPC operation; for example, for all the cases of $N_w = 2, 4$ and 8, the average phase error is reported as ~1.4% when using $N = 0.5 N_i N_o N_w$, which reduced to ~1.0% when using $N = 2 N_i N_o N_w$. Exemplary multi-wavelength output fields from the diffractive OPC model using $N = 2 N_i N_o N_w$ and $N_w = 8$ wavelengths are provided in Supplementary

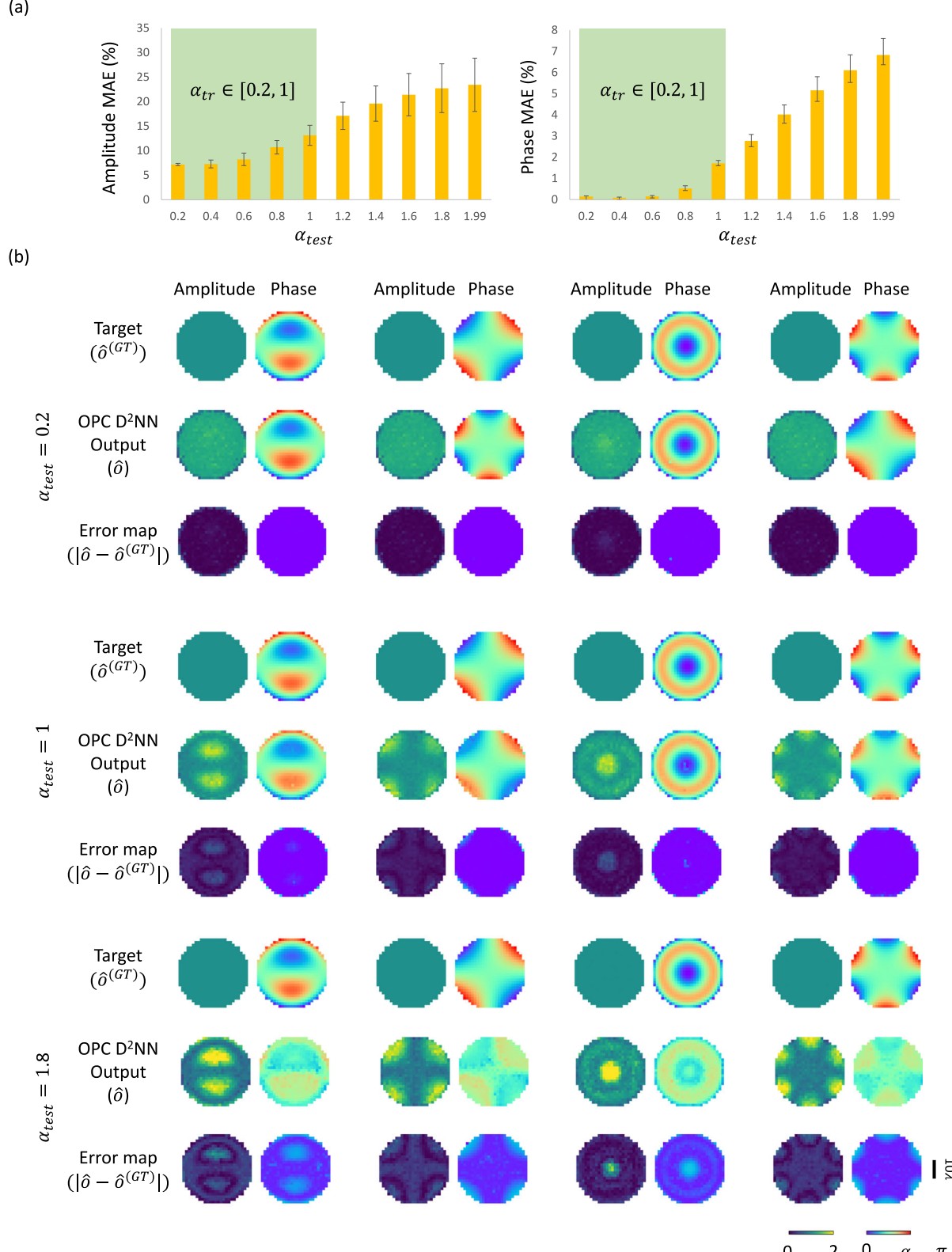

**Fig. 3 | The impact of the phase contrast of aberrated input fields on the performance of OPC using the diffractive processor design shown in Fig. 2b.**
**a** Amplitude and phase MAE values between the diffractive OPC outputs and their ground truth as a function of the input phase contrast parameter $\alpha_{test}$. Metrics are benchmarked across the dataset, reported as mean values with SDs shown as error bars. This quantification is performed utilizing the same testing set as in Fig. 2d. $\alpha_{test}$ indicates that the input complex field has a dynamic phase range of $[0, \alpha_{test}\pi]$. **b** Exemplary visualization of the output complex fields all-optically synthesized by the diffractive OPC processor when using phase aberrated input fields with different phase contrast $\alpha_{test}$, along with their ground truth and absolute error maps.

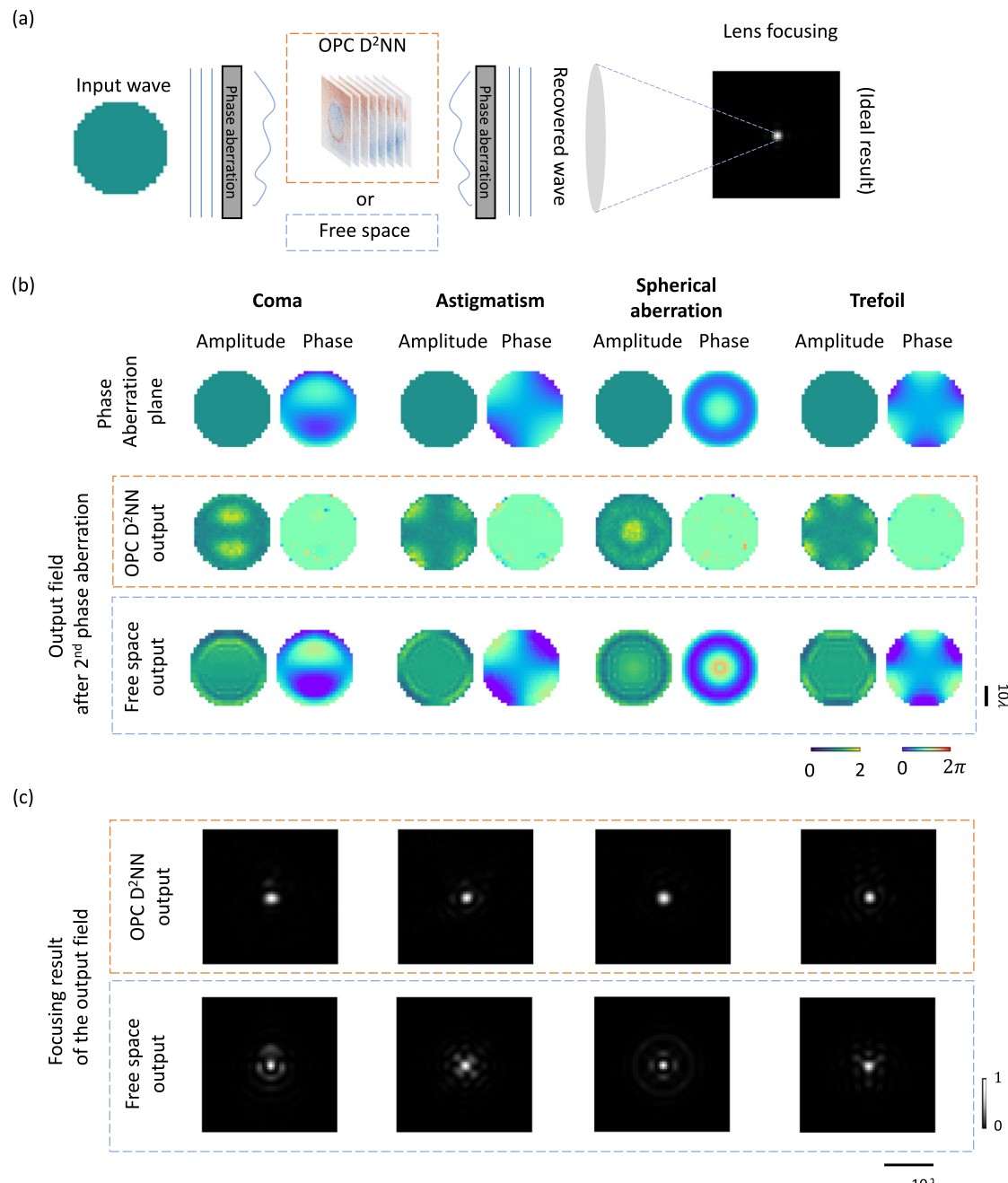

**Fig. 4 | Simulation results for correcting phase aberration-induced wavefront distortions using the diffractive OPC design shown in Fig. 2b. a** Illustration of a scenario that leverages the diffractive OPC processor to correct phase aberration-induced wavefront perturbations. The random, unknown phase perturbations induced before and after the diffractive processor are identical to each other. **b** Examples of the output complex fields immediately after the second phase perturbation plane. **c** Intensity distributions obtained by focusing the output fields in (**b**) through a diffraction-limited lens.

Fig. S3a, demonstrating a decent match with their ground truth fields across all the eight wavelength channels. We also tested the same diffractive OPC model with structurally more complex input fields composed of 3 random Zernike polynomials (never used in the training); as shown in Supplementary Fig. S3b, the diffractive processor still revealed output phase profiles that closely align with their ground truth across all the 8 wavelength channels. All these blind testing results corroborate our framework's feasibility for performing multi-wavelength OPC operations at different parts of the spectrum.

These multi-wavelength diffractive OPC designs shown on the left part of Supplementary Fig. S2 employed N-BK7 as the diffractive layer material, which has a known dispersion curve with varying refractive

index values as a function of the wavelength, which was numerically modeled in our diffractive designs. Next, we investigated the feasibility of constructing a broadband diffractive OPC processor using a material with a different dispersion property; to highlight an extreme case, we assumed a diffractive material with *flat dispersion* such that the refractive index does not change as a function of the wavelength within the operation band of interest. We selected this dispersion-free material to highlight an important feature of multispectral diffractive OPC designs: their phase conjugation performance at different wavelengths is independent of the material dispersion. Therefore, in this analysis, we adopted a diffractive material with flat dispersion, exhibiting a constant refractive index ($n = 1.7$) within its operational band

(400 nm–750 nm). As reported in the right part of Supplementary Fig. S2, these dispersion-free OPC diffractive designs achieve average phase errors of 1.34% and 0.95% across $N_w = 8$ wavelength channels when using $N = 0.5 N_i N_o N_w$ and $2 N_i N_o N_w$, respectively. Remarkably, they attain performance parity with previous designs that utilized N-BK7, even exhibiting a slight improvement. The output examples of the dispersion-free OPC processor design using $N = 2 N_i N_o N_w$ and $N_w = 8$ wavelengths are also provided in Supplementary Fig. S4, all presenting very good agreement with the ground truth. These findings confirm that the broadband OPC capability of our diffractive processors is not subject to specific dispersion properties of the materials, thereby highlighting it as a versatile platform that can adapt to varying material choices and operation wavelengths.

## Experimental validation of transmissive diffractive OPC networks

Next, we sought to experimentally validate our diffractive OPC framework using a set-up based on monochromatic terahertz illumination. As illustrated in Fig. 5a, the objective here is to use a diffractive OPC processor to perform phase conjugation of a wavefront, which is emitted by a pinhole-like object and subsequently distorted by a random, unknown phase perturbation plane. The resulting phase-conjugated wavefront (by the diffractive OPC processor) passes through another identical phase perturbation plane to be refocused onto a small point – if the OPC was successful. In this experimental configuration, we used a diffractive OPC design comprising $K = 3$ phase-only dielectric diffractive layers ($L_1$–$L_3$), positioned between two identical random, unknown phase perturbation planes with a phase contrast parameter of $\alpha_{test} = 0.5$. The input pinhole object is set as a square-shaped aperture with a size of $4.3\lambda$, situated at a distance $d$ before the first random phase perturbation plane. The wavefront exiting the diffractive OPC system is expected to focus on a spot of the same size as the input object at the same distance $d$ after the second phase perturbation plane. The intensity distribution of this focusing spot is measured at the output plane, with its quality serving as the figure-of-merit of this experimental OPC validation. The detailed structural parameters used for this experimental design are reported in Fig. 5a and the Methods section.

It is worth noting that, compared to the optical configuration used in the prior subsection (i.e., Fig. 4a), the experimental configuration in Fig. 5a possesses some differences. The main difference lies in that, in the experimental configuration, the optical field impinging onto the first phase perturbation plane has a spherical wavefront emitted by a pinhole, rather than a uniform plane wave as the one in Fig. 5a. Therefore, the complex field $i$ that enters the diffractive OPC system's aperture after propagating through the first phase perturbation plane and the expected (ground truth) output complex field $o^{(GT)}$ that exits from the diffractive OPC system before the second phase perturbation plane can be written as:

$$i(x,y) = s(x,y)e^{j\Psi(x,y)} \quad (3)$$

$$o^{(GT)}(x,y) = i^*(x,y) = s^*(x,y)e^{-j\Psi(x,y)} \quad (4)$$

Here in Eqs. (3) and (4), $\Psi$ denotes the phase profile of the randomly generated, unknown phase perturbation plane, and $s$ represents the illumination field at the phase perturbation plane, emitted from the pinhole object. In our experimental set-up, due to the spherical wave characteristics of $s(x,y)$, both its amplitude and phase distributions exhibit an increasingly rapid change from the center outwards to the edges. This leads to a significantly larger phase contrast $\alpha_{test}$ in the actual input field. Therefore, if the diffractive OPC processor can successfully retrieve (at the output plane) the image of the input (pinhole) object, despite the presence of the randomly

generated phase aberrations, it would indicate the successful implementation of the diffractive system's phase conjugation capability.

To train our experimental OPC design using $K = 3$ diffractive layers, we generated a set of random phase perturbation planes and optimized our experimental diffractive OPC design through deep learning. The phase profiles of the resulting three diffractive layers are visualized in the left column of Fig. 5b. We fabricated these diffractive layers using 3D printing, with the photos of the fabricated layers presented in the right column of Fig. 5b. After assembling these diffractive layers, we employed a THz source ($\lambda = 0.75$ mm) and detector to measure the resulting intensity distribution at the output plane. The schematic and photographs of the experimental set-up are presented in Fig. 5c and d.

During the experiments, we tested our system using four different phase perturbation planes (also fabricated through 3D printing), which were never seen during the training ($\alpha_{test} = 0.5$). Figure 6 shows the experimental measurements, which successfully revealed a Gaussian-shaped circular spot pattern, closely aligning with the corresponding numerical simulation results. Furthermore, we also tested the system without placing the diffractive OPC network between the two-phase perturbation planes, i.e., with only free space propagation. The resulting experimental images displayed only scattered patterns, starkly contrasting with the results achieved using the diffractive OPC processor.

In addition to these experimental results, we also performed an ablation study by constructing several baseline configurations and comparing their performances. As shown in Supplementary Fig. S5a, these baseline configurations include replacing the three-layer diffractive OPC processor with (1) a single-layer diffractive OPC processor, (2) a conventional thin lens, and (3) free space. Here, we used the same phase perturbation planes as in the experiments, but adjusted the phase contrast to $\alpha_{test} = 1$, which sets a more challenging test. After the training, the resulting thickness profiles of the diffractive layers for these baseline configurations are provided in Supplementary Fig. S5b, and their test results are compared in Supplementary Fig. S6. From these comparative analyses, it can be found that the single-lens and the single-diffractive-layer systems produce similar results: while they can both generate a spot at the output plane, the positions of the spots do not appear accurately at the center, revealing beam focusing artifacts and these spots are accompanied by various noise patterns at the output plane. The results of the $K = 3$ diffractive OPC system, on the other hand, provide superior results to both of these baseline systems, as shown in Supplementary Fig. S6. These observations also align with the architectural depth advantages of diffractive visual processors, demonstrated for various applications by achieving better approximation accuracy for a given task when the diffractive degrees of freedom are distributed across a deeper architecture[28,32,33,35,53].

Apart from the monochromatic diffractive OPC design reported above, we also performed experimental validation of a multi-wavelength OPC processor. Following the same architecture and training method used by the monochromatic model shown in Fig. 5, we trained a diffractive multi-wavelength OPC processor to simultaneously support phase conjugation operation at three distinct wavelengths: 0.75, 0.775, and 0.8 mm. The resulting diffractive layers were fabricated using 3D printing, as shown in Fig. 7a. In this experiment, we employed the same input aperture as in the previous test and sequentially illuminated the phase perturbation plane with each operational wavelength, capturing the corresponding intensity distribution within the output aperture. The experimental results, presented in Fig. 7b, show that our diffractive multi-wavelength OPC processor consistently generated Gaussian-shaped circular output patterns for all three wavelengths, achieving similar results to its monochromatic counterpart shown in Fig. 6b. This success further substantiates the feasibility of designing diffractive OPC processors capable of correcting broadband aberrated fields across multiple

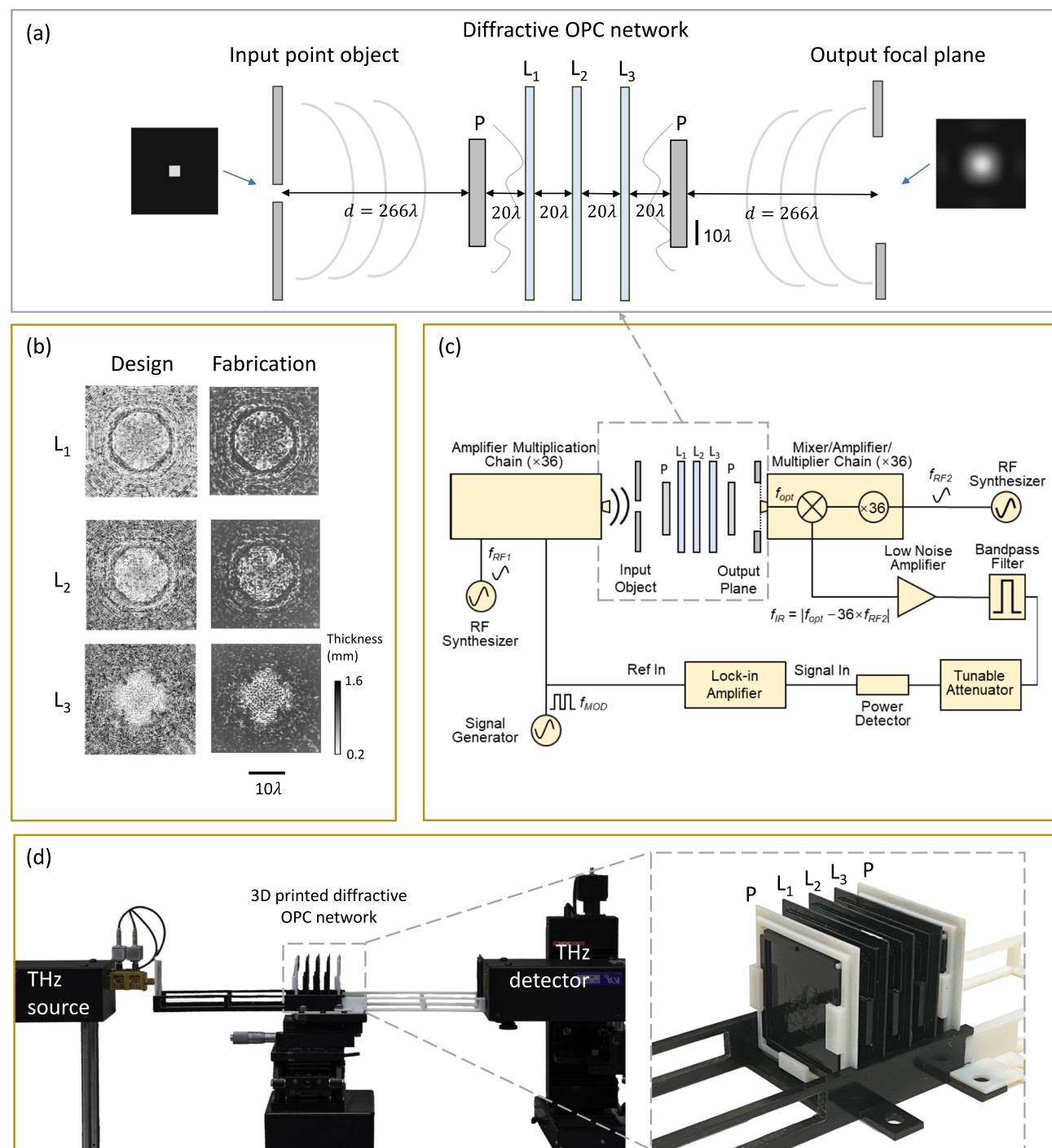

**Fig. 5 | Experimental set-up of the transmissive diffractive OPC processor.**
**a** Illustration of a diffractive OPC processor composed of three diffractive layers ($L_1$, $L_2$, $L_3$) to perform OPC operation on a diverging wavefront, emitted by a pinhole-like object and subsequently distorted by a random, unknown phase perturbation plane (P) – which was never seen during the training process. The second phase perturbation plane positioned after the diffractive OPC processor is identical to the first one to test the efficacy of the phase conjugation operation. **b** Thickness profiles of the trained diffractive layers (left column) and the photographs of their fabricated versions using 3D printing (right column). **c** Schematic of the terahertz imaging set-up. **d** Photographs of the experimental set-up, including the fabricated diffractive OPC processor (inset).

wavelength channels, with an independent aberration in each channel.

## Output power efficiency of diffractive OPC designs
In traditional AOPC solutions, the output power of the phase-conjugated beam could be weak due to the nonlinear optical processes involved in AOPC systems, often leading to a low power efficiency of <1%. While the digital OPC methods can provide considerable power due to incorporating active illumination and modulation in their playback process, they present other limitations, including increased system complexity and reduced operation speed. For our diffractive OPC framework, its power efficiency is directly associated with the output diffraction efficiency exhibited by the diffractive volume. Based on theoretical evidence and numerical studies

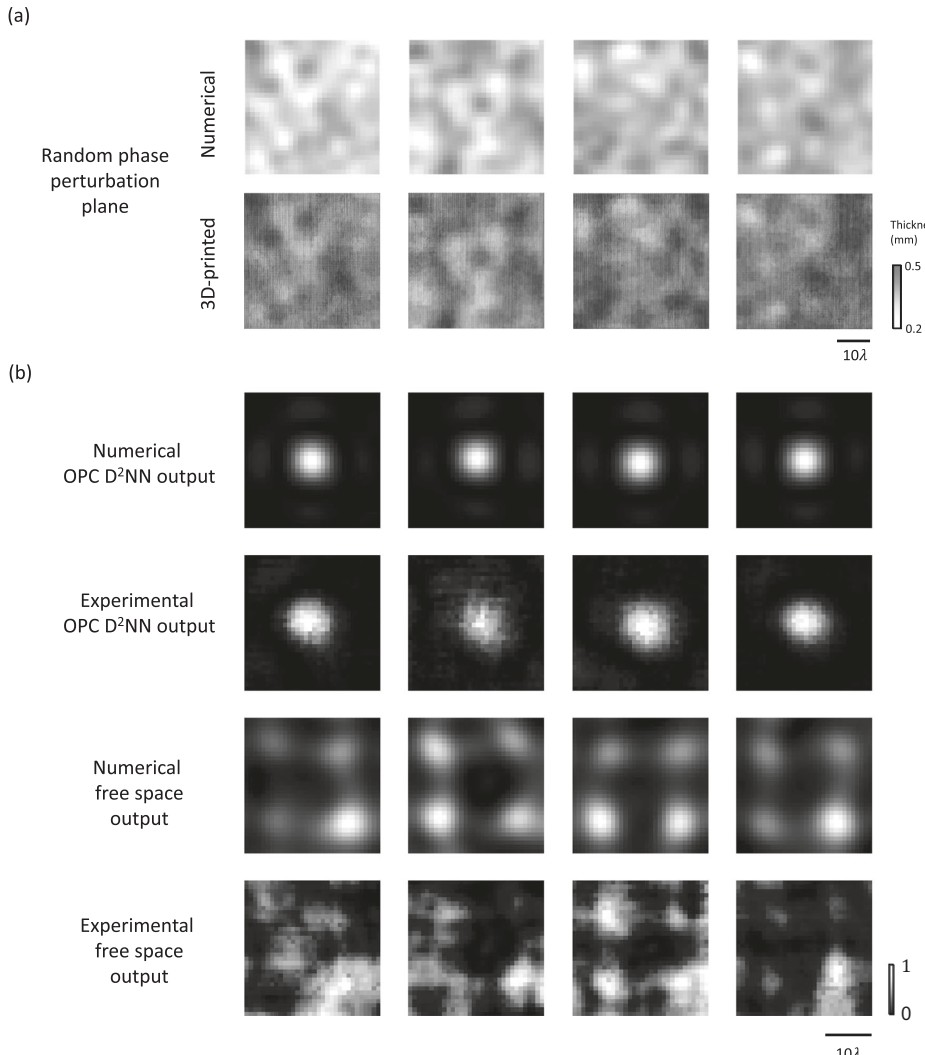

**Fig. 6 | Experimental results of the transmissive diffractive OPC design shown in Fig. 5b. a** Visualization of the phase perturbation planes, along with the photographs of their fabricated versions. **b** Numerically simulated and experimentally measured intensity distributions at the output plane, compared with the free-space output results in the absence of the diffractive OPC processor.

presented in earlier works[33,53,54], axially deeper diffractive processor architectures were proven advantageous in terms of their transformation accuracy and output diffraction efficiency. Here, we analyzed the impact of the number of trainable diffractive layers ($K$) on the OPC performance quantified by the output phase MAE. Taking our design shown in Fig. 2b with $K = 8$ as the baseline design for this analysis, we trained its counterparts with different numbers of diffractive layers, e.g., $K = 4$, 6, and 10, by maintaining the other structural parameters identical to the baseline design, also using the same training dataset. Figure 8a reports these designs' resulting output phase MAE values as a function of $K$. It is evident that as $K$ increases the phase conjugation errors of the diffractive output fields decrease; for example, the phase MAE values at $K = 4$ and 10 are reported as $2.37 \pm 0.37\%$ and $1.55 \pm 0.21\%$, respectively. These findings are also confirmed through the exemplary visualization results shown in Fig. 8b, where the output phase profiles from the models with $K = 8$ and 10 present much better similarity to their ground truth when compared to the models with smaller $K$. Furthermore, the output diffraction efficiencies of these OPC designs also present an increasing trend as more diffractive layers are used. For instance, the model with $K = 4$ yields an output diffraction efficiency of $13.8 \pm 1.2\%$, and this efficiency increases to $24.4 \pm 1.2\%$ when $K = 10$ is used, marking an efficiency increase of 10.6%. These diffraction efficiency metrics underscore that a deeper architecture for

our diffractive processors achieves both better OPC performance and better output power efficiency by exploiting the larger degrees of freedom and the depth advantage.

Although the diffractive OPC designs shown in Fig. 8 all present output diffraction efficiencies of > 13–25%, OPC systems that are even more power-efficient might be sought for various real-world applications, involving e.g., low signal-to-noise ratio image sensors. To address this need, one can introduce an additional diffraction efficiency-related loss term[34,35,43,44,54,55] to the training loss function, aiming to balance the tradeoff between the OPC performance and the diffraction efficiency of the diffractive processor (see the "Methods" for details). This strategy was already employed in the experimental design shown in Fig. 5b to enhance the diffraction efficiency of the output signals. In Fig. 9, we present an in-depth quantitative exploration of this tradeoff relationship between the OPC performance and the output diffraction efficiency. For this analysis, we revisited the two designs with $K = 4$ and 8 shown in Fig. 8; these designs respectively exhibit diffraction efficiencies of $13.8 \pm 1.2\%$ and $20.2 \pm 0.9\%$, with phase MAE values of $2.37 \pm 0.37\%$ and $1.71 \pm 0.21\%$, and amplitude MAE values of $10.80 \pm 2.13\%$ and $10.80 \pm 2.84\%$. Keeping the structural parameters identical as before and utilizing the same training/testing dataset, we retrained these diffractive OPC designs (from scratch) by applying varying degrees of diffraction efficiency penalty to the

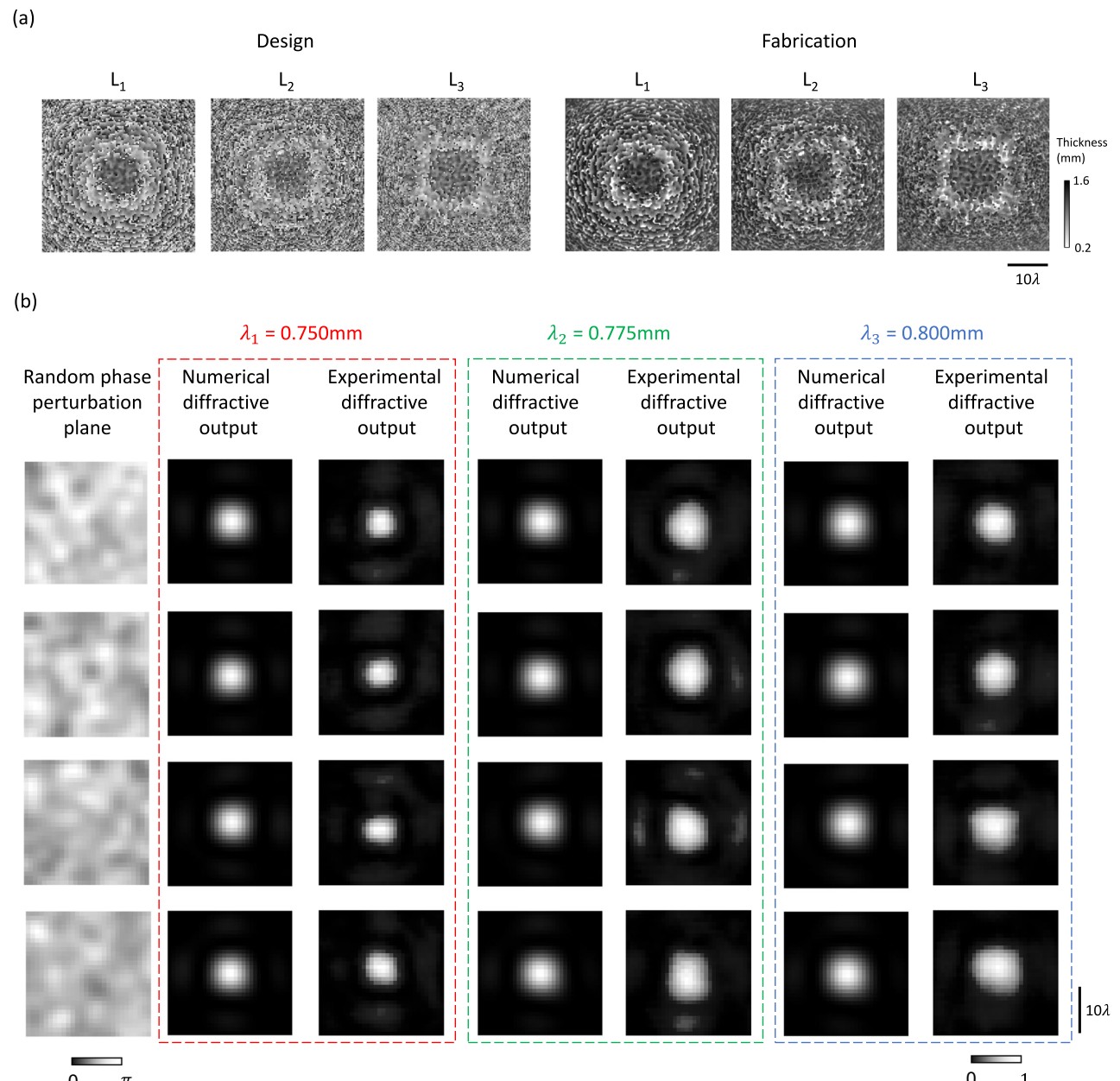

**Fig. 7 | Experimental results of the transmissive diffractive multi-wavelength OPC processor design. a** Thickness profiles of the trained diffractive layers (left) and the photographs of their fabricated versions using 3D printing (right). **b** Phase perturbation planes used in the experiments, along with their resulting output multi-wavelength intensity distributions obtained from numerical simulations and experimental measurements.

training loss functions, resulting in new designs with enhanced output diffraction efficiencies. Figure 9a depicts the phase and amplitude MAE values of these new designs in relation to their output diffraction efficiencies. These results reveal that, compared to the original $K = 4$ design shown in Fig. 8, the newly trained diffractive designs achieved a ~ 6-fold increase in their output diffraction efficiencies, reaching up to 85.9 ± 1.7%. This improvement comes with a modest sacrifice in their phase conjugation performance, with phase MAE values reaching up to 3.23 ± 0.38%. Similarly, compared to the original $K = 8$ design shown in Fig. 8, these new designs subjected to the diffraction efficiency penalty showcased a further improved diffraction efficiency of up to 92.1 ± 1.5%, with only marginal surges in the phase MAE values that reach up to 2.26 ± 0.32%.

In these analyses, we also observed an intriguing phenomenon: as the diffraction efficiency increases, there is a slight decrease in the

amplitude MAE values of the diffractive OPC designs. These results suggest that a stronger diffraction efficiency-related penalty term results in a greater portion of the lower spatial frequency modes being directed into the output aperture, leading to a more uniform output field amplitude distribution, which lowers the amplitude MAE values. Figure 9b offers a visual representation of these designs' diffractive output fields, further corroborating our findings. Collectively, these results suggest that diffractive OPC processors can provide a favorable balance between their phase conjugation performance and output diffraction efficiency, which can be improved as desired by using an appropriate training loss function.

**Diffractive phase-conjugate mirror design**

In our results and analyses presented so far, we utilized diffractive OPC processors in a transmission geometry. While the transmissive

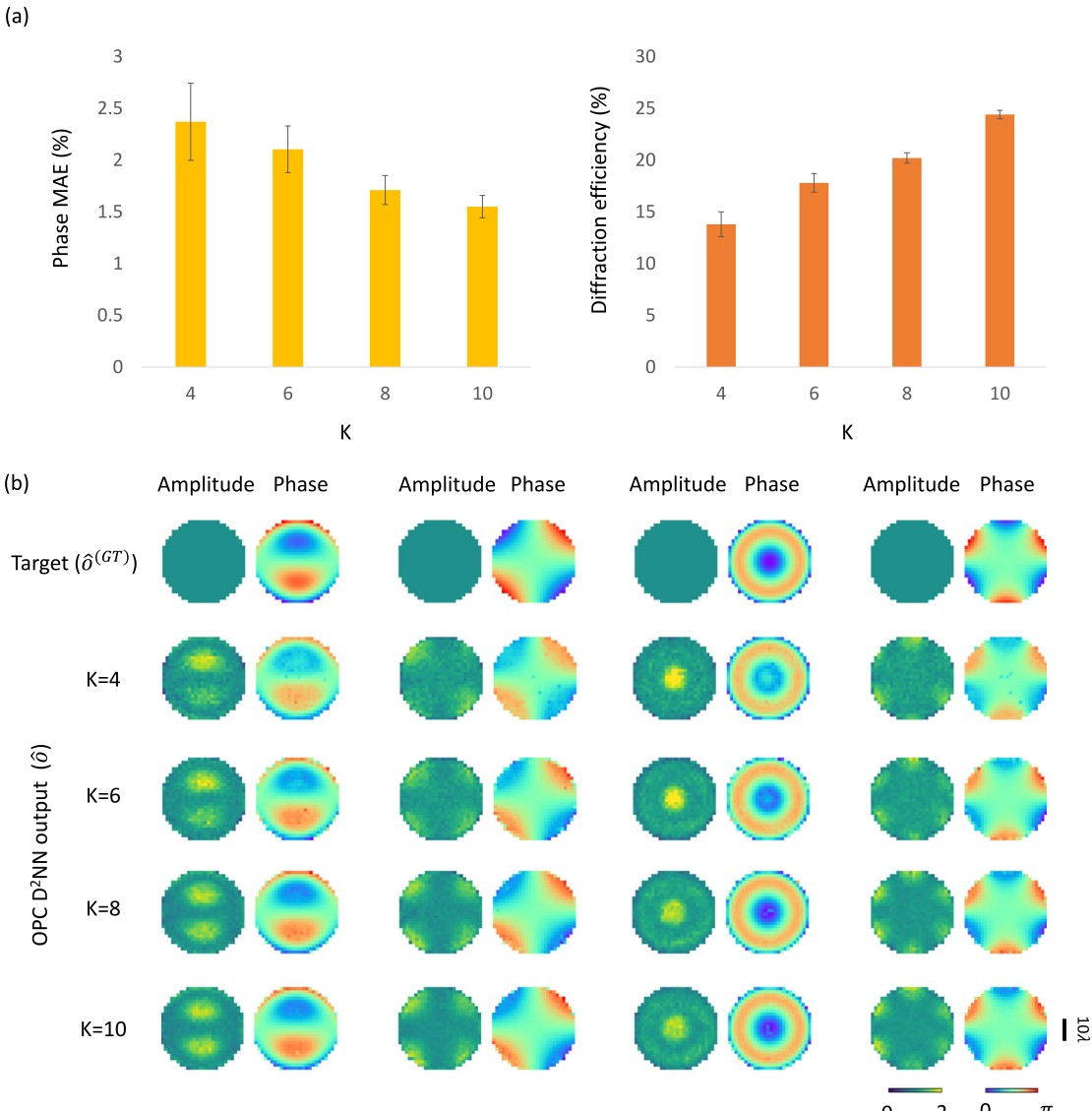

**Fig. 8 | The impact of the number of diffractive layers on the phase conjugation performance and the output diffraction efficiency of the diffractive OPC processors. a** Phase MAE values and the output diffraction efficiencies of the diffractive OPC outputs as a function of the number of layers ($K$) used in the diffractive OPC processor design. Metrics are benchmarked across the dataset, reported as mean values with SDs shown as error bars. **b** Exemplary visualization of the diffractive output fields produced by various diffractive OPC processor designs with different $K$ values (4 –10).

structure we illustrated can be retrofitted into a reflective OPC configuration, this would complicate the system with additional optical components such as beam splitters and image projection systems. To create a simpler solution for designing a phase-conjugate mirror, we merged a diffractive OPC system with an ordinary reflective mirror, which creates a double-pass configuration through the same diffractive layers to all-optically perform OPC in reflection mode. As illustrated in Fig. 10a, the incident phase aberrated field, denoted as $i = e^{j\psi}$, first propagates through the diffractive layers ($L_1, ..., L_K$), resulting in an intermediate field $o^{(\text{intermediate})} = \text{D}^2\text{NN}_{\text{forward}}\{e^{j\psi}\}$. Upon reflection from the planar mirror, this reflected intermediate field (i.e., $Ro^{(\text{intermediate})}$) traces its path in the reverse direction across the same set of diffractive layers, ultimately collected in reflection through the same aperture used for input. This produces a final reflected output field $o = \text{D}^2\text{NN}_{\text{backward}}\{Ro^{(\text{intermediate})}\}$, i.e., $o \approx \alpha \cdot o^{(\text{GT})} = \alpha \cdot i^* = \alpha \cdot e^{-j\psi}$.

Following this optical configuration, we numerically modeled a diffractive phase-conjugate mirror that shares the same set of design parameters as the previous transmissive OPC design presented in Fig. 2a. Utilizing the same dataset used by the analysis reported in

Fig. 2, we trained and validated our diffractive model, with the resulting phase profiles of the diffractive layers shown in Supplementary Fig. S7. Our numerical simulations achieved average phase and amplitude MAE values across different test sets: for input phase profiles constituted by two randomly selected Zernike polynomials, the phase and amplitude MAE values were $1.25 \pm 0.05\%$ and $9.41 \pm 2.87\%$, respectively; for those input fields using a single randomly selected Zernike polynomial, the MAE values were $1.31 \pm 0.07\%$ and $11.10 \pm 3.93\%$, respectively; and for those featuring three randomly selected Zernike polynomials, the MAE values were $1.25 \pm 0.04\%$ and $7.92 \pm 2.20\%$, respectively. Moreover, the visual illustrations reported in Fig. 10c–e revealed that the diffractive output fields present a very good agreement with their corresponding ground truth phase profiles, confirming the success of the OPC operation. The amplitude distributions of the reflected fields also generally retain uniformity with minor errors that stand at a similar level as the previous transmissive OPC designs. Overall, these results affirm the feasibility of designing a diffractive OPC processor coupled with an ordinary mirror to create a reflective phase-conjugate mirror, which might find various

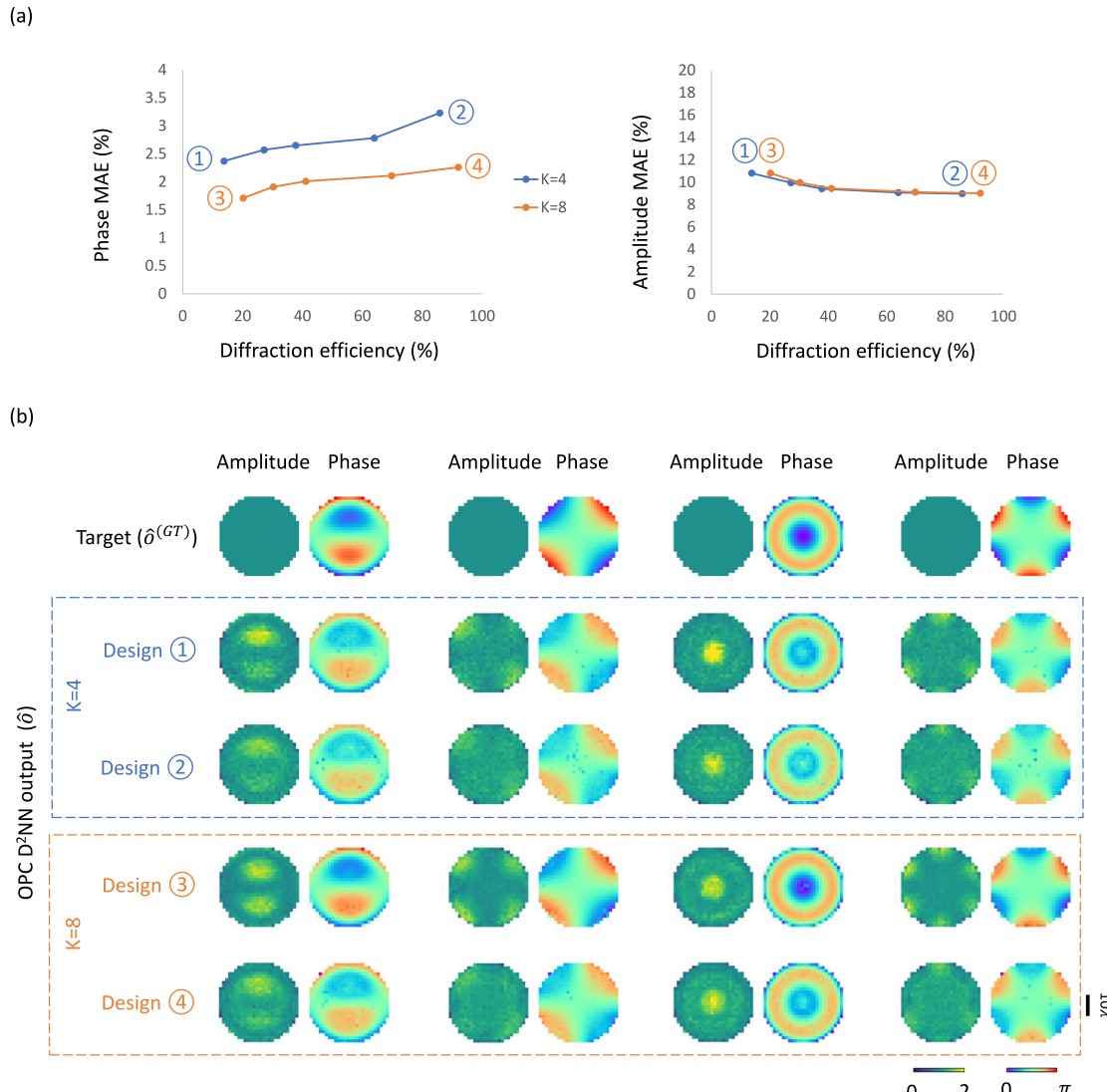

**Fig. 9 | Analysis of the tradeoff between the all-optical phase conjugation performance and the output diffraction efficiency of the diffractive OPC processors. a** The phase MAE values (left) and the amplitude MAE values (right) of the diffractive OPC design output fields with various levels of diffraction efficiency penalty, plotted as a function of the output diffraction efficiencies. Two sets of diffractive OPC designs using $K = 4$ and 8 diffractive layers were trained and blindly tested. Specifically, ① and ② depict different 4-layer diffractive designs resulting from the use of $\beta_{Eff} = 0$ and $\beta_{Eff} = 1$, respectively, which refer to the weight of the diffraction efficiency-related penalty in the training loss function (see Eq. (18)). ③ and ④ represent the counterparts of ① and ② for the designs with $K = 8$ diffractive layers. **b** Exemplary visualization of the diffractive output fields produced by various diffractive OPC processor designs with different $K$ values (4 and 8) and various levels of diffraction efficiency-related penalty terms.

applications in e.g., turbidity suppression and atmospheric aberration correction, among many others.

## Discussion

In our Results section, we successfully demonstrated the generation of phase-conjugated output fields using diffractive OPC processors. Nonetheless, some imperfections can be observed in the output amplitude profiles, especially in the reflective design of the diffractive phase-conjugate mirror. These non-uniformity-related errors in the output amplitude profiles can be mitigated and suppressed by adjusting the hyperparameter that governs the amplitude uniformity penalty in our training loss function (see the Methods section for details). To highlight this opportunity, we provide further analysis of this amplitude non-uniformity in Fig. 10b based on the transmissive and reflective designs presented in Fig. 2b and Supplementary Fig. S7, respectively, utilizing input fields composed of two randomly selected Zernike polynomials. By changing the penalty threshold (V) associated

with the level of non-uniformity in the output amplitude field, we designed different OPC processors and plotted the corresponding changes in the average phase MAE values against the average amplitude MAE values. Notably, for the transmissive design, the amplitude error decreased from 8.89% ± 2.04% to 4.51% ± 0.76%, alongside a relatively small phase error increase from 1.38 ± 0.12% to 1.97 ± 0.15%. A similar pattern is also observed for the reflective OPC designs, with the amplitude error dropping from 9.41% ± 2.23% to 4.60% ± 0.78% and the phase error rising from 1.25 ± 0.11% to 1.67 ± 0.13%. These results indicate that more uniform amplitude profiles can be achieved at the output aperture of our diffractive OPC processors with only a minor compromise in output phase accuracy. Moreover, our analyses suggest that reflective designs outperform their transmissive counterparts in terms of both amplitude and phase errors, which can be attributed to their double-pass configuration that more efficiently exploits the available degrees of freedom within the diffractive volume. These findings are further supported by the visualization

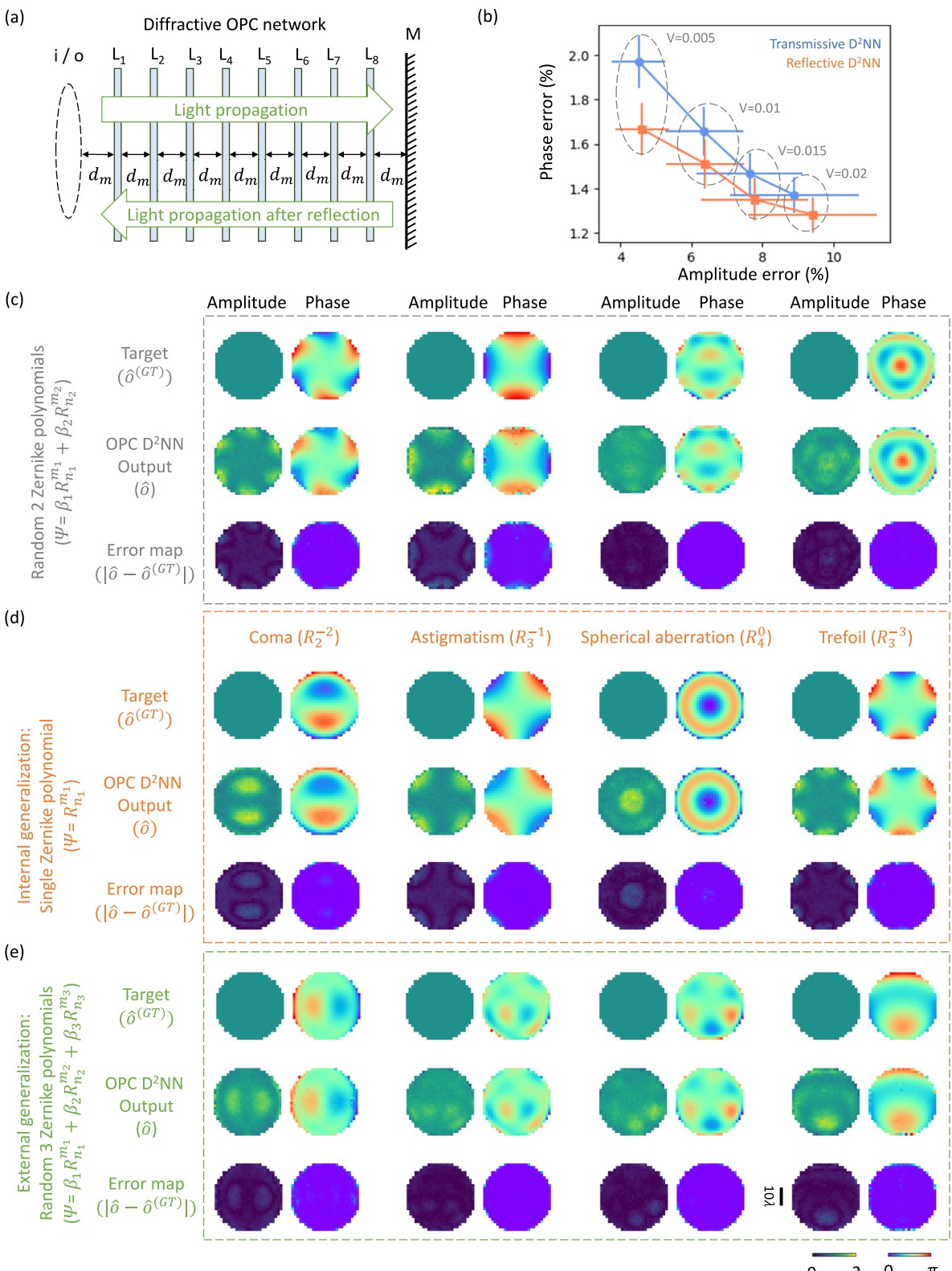

**Fig. 10 | The diffractive phase-conjugate mirror design and the visualization of the output field examples. a** Illustration of a diffractive OPC processor operating in reflection mode, together with a standard mirror, forming a diffractive *phase-conjugate mirror*. **b** Comparison of the phase error values and the amplitude error values between reflective OPC designs and transmissive OPC designs using different amplitude MSE thresholds (V) during their training/design. Metrics are benchmarked across the dataset, reported as mean values with SDs shown as error bars. **c**–**e** Same as Fig. 2c–e except for using the diffractive phase-conjugate mirror design.

examples provided in Supplementary Fig. S8, showing enhanced amplitude uniformity when the amplitude penalty threshold (V) is adjusted.

To better understand our diffractive OPC models' generalization capability, we deliberately increased the complexity of the test data by generating input phase profiles with ≥3 Zernike polynomials, introducing phase structures that are spatially more complex, exhibiting rapid variations. This blind testing evaluation was conducted using the same diffractive phase-conjugate mirror model that was originally trained on phase profiles constructed from two randomly selected Zernike polynomials. The results of this evaluation, demonstrated in Supplementary Fig. S9a, reveal that our diffractive OPC processor maintained high accuracy levels, with phase errors remaining < 1.3% across input datasets incorporating 3 to 8 randomly selected Zernike polynomials. Furthermore, the examples of the output fields, depicted in Supplementary Fig. S9b–d, showcase very good concordance with their corresponding targets. These results highlight the OPC processor's versatility and effectiveness, affirming its capability to accurately handle a broad range of wavefront patterns, including various highly structured phase distortions and aberrations (never seen before). This underscores the utility of the presented platform as a general-purpose optical phase conjugation device.

In the practical implementations of our diffractive OPC processors, mechanical misalignment between different elements could constitute a notable challenge to their phase conjugation performance, as they can cause the optical waves to be modulated by the diffractive layers in an undesired way, leading to results deviating from their designed performance. To provide further evidence for this, we utilized the diffractive OPC processor model previously shown in Fig. 2a and subjected the diffractive layers to different magnitudes of random displacements, either in the lateral directions ($\Delta_x, \Delta_y \in U[-\Delta_{xy,\text{test}}, \Delta_{xy,\text{test}}]$) or the axial direction ($\Delta_z \in U[-\Delta_{z,\text{tr}}, \Delta_{z,\text{tr}}]$), sampled from random uniform distributions ($U$). The resulting OPC performances of these misaligned diffractive processors are summarized in Supplementary Figs. S10 and S12 using blue curves, which reveal a clear trend of increasing degradation in both the amplitude and phase errors as the misalignment gets more severe. To address this misalignment challenge, a "vaccination" strategy can be applied during the training process by modeling these misalignment errors as random noise into the numerical forward model of the system[34,37,56]. Specifically, the 3D random displacements of the diffractive layers ($\Delta_x$, $\Delta_y$ and $\Delta_z$) can be modeled using random variables, changing from iteration to iteration during the training process, to provide substantial resilience against such random displacements at a tolerable cost of performance loss. The efficacy of this vaccination-based design strategy is demonstrated in Supplementary Figs. S10 and S12, where new diffractive OPC processor models were trained under random lateral and axial misalignments of varying magnitudes. These vaccinated diffractive OPC processor results (shown in orange and green curves) reveal that if the training parameters $\Delta_{xy,\text{tr}}$ and $\Delta_{z,\text{tr}}$ encompass the range of misalignments encountered in the blind testing ($\Delta_{xy,\text{test}}$ and $\Delta_{z,\text{test}}$), the impact of such physical/random misalignments on OPC performance can be significantly reduced. For instance, compared to the baseline model that was not "vaccinated" against these misalignments, the average output phase error of the "vaccinated" model trained using $\Delta_{xy,\text{tr}} = 0.24\lambda$ is reduced to 3.30% when tested with $\Delta_{xy,\text{test}} = 0.24\lambda$. Visual analyses reported in Supplementary Fig. S11 and S13 further corroborate these findings, showing that the diffractive processor maintains high OPC output accuracy under different levels of misalignments, even when using structurally more complex aberrated input fields that are formed by a greater number of Zernike polynomials, as illustrated in Supplementary Figs. S11b and S13b. These results highlight our diffractive OPC processor's capability to accommodate various phase aberration patterns and withstand unexpected random 3D misalignments among the diffractive layers. It is also possible to adopt the same vaccination strategy to mitigate the impact of rotational misalignments between the diffractive layers when such misalignments become a critical factor in experiments. Specifically, in-plane rotations of the diffractive layers can be modeled with 2D coordinate transformations based on unitary rotation matrices, and the out-of-plane rotations can be accounted for by adjusting the wave propagation forward model between randomly tilted diffractive planes[56–58]. Beyond such mechanical misalignments in 3D, this vaccination strategy can also be extended to counteract other types of potential errors, such as fabrication imperfections in diffractive layers, inaccuracies in material dispersion characterization, and detection noise, thereby enhancing the practical robustness of the diffractive OPC system.

The presented diffractive OPC processor designs offer significant advantages with their multi-wavelength operation, distinctly setting them apart from conventional OPC methods. While conventional AOPC systems are typically limited to narrowband operation due to their dependency on wavelength-specific nonlinear coefficients of materials, our diffractive OPC processors employ basic dielectric materials without the need for a specific dispersion relationship (see Supplementary Fig. S2). Our diffractive designs allow for multi-wavelength OPC operations, facilitating efficient performance over a wider spectral range, independent from the dispersion characteristics of the diffractive materials. For example, we showcased a broadband diffractive OPC processor designed using N-BK7, a widely used borosilicate glass known for its high damage threshold and superior optical quality, making it well-suited for a vast spectrum of wavelengths in the visible. In applications that demand a higher damage threshold, fused silica emerges as a viable alternative for the diffractive layer material, with UV to near-infrared band transmission and negligible thermal expansion[59].

Another key benefit of our framework is its exceptionally fast response time, performing OPC at the speed of light propagation, a feature paramount for numerous wavefront shaping (WFS) applications. An optimal WFS technique for scattering suppression would need to combine rapid system response, power efficient generation of the conjugated wave, and extensive degrees of freedom for precise wave manipulation[15,60]. Speed is especially crucial for WFS, as it must be accomplished within the speckle correlation period, typically < 1 ms, to be compatible with the dynamic nature of various specimens, such as living organisms and tissue samples. This requirement poses a challenge for conventional DOPC methods due to inherent delays in their operation. In contrast, the presented diffractive OPC processor, being entirely optical, achieves OPC operation as the light is transmitted through a thin optical volume, thus effortlessly satisfying the crucial demand for rapid response. To shed more light on this, we calculated the optical response time of our transmissive OPC processors following the same design illustrated in Fig. 2b. Here, the response time is determined by the light propagation time throughout the entire diffractive volume (from the input plane to the output plane), encompassing all the free space propagation between the layers as well as within the diffractive layer materials. Specifically, with an axial spacing of 12λ between successive layers and a base thickness of 0.2 mm for each diffractive layer, the OPC response times can be calculated as ~293 ps and ~123 ps at $\lambda = 0.75$ mm and $\lambda = 0.3$ mm, respectively. For our diffractive phase conjugate mirror designs, which entail a double-pass configuration, the optical response times will be doubled compared to their transmissive counterparts. For diffractive OPC designs operating at the visible and IR part of the spectrum, for example, the optical response times will be further reduced by more than one order of magnitude as the thickness of the diffractive processor volume can be significantly reduced due to the shorter wavelength. This significant speed advantage compared to existing AOPC and DOPC systems reported in the literature[10,12,15] underscores the utility of our diffractive designs for applications that necessitate ultra-

fast OPC response. Furthermore, by incorporating a diffraction efficiency penalty in the training loss function and enhancing the system's degrees of freedom through an increased number of diffractive features, our method holds the potential of achieving even higher power efficiency (see e.g., Figs. 8–9), also broadening the scope of operation beyond the limits of traditional OPC methods. These advantages render our framework particularly well suited for applications in biomedical imaging and potentially for astronomical observations through scattering media such as the atmosphere, where energy efficiency is critically important[61].

In summary, the results and analyses of this manuscript have successfully demonstrated diffractive OPC processors that can all-optically perform phase conjugation of input complex fields with arbitrary unknown phase distributions, without the need for any digital computation, image acquisition, or active beam modulation. Through simulations, the accuracy of our diffractive OPC operation was analyzed, revealing the empirical relationship between the phase conjugation accuracy and the phase contrast of the input fields. Furthermore, we experimentally validated this concept at the terahertz part of the spectrum by 3D fabricating a diffractive OPC processor, which successfully processed randomly generated phase-aberrated input fields. We also demonstrated that the presented diffractive OPC processor designs can be scaled (expanded/shrunk) to extend their operation to other parts of the electromagnetic spectrum, including the visible and IR bands. Such diffractive OPC processors operating at shorter wavelengths can be fabricated using appropriate nano-/micro-fabrication methods, such as two-photon polymerization-based 3D fabrication[62–64].

## Methods

### Optical forward model of the diffractive OPC processors

To model a diffractive OPC processor, its diffractive layers are treated as thin planar elements that modulate the complex field of the incident coherent light. For the $q^{th}$ diffractive feature of the $l^{th}$ layer positioned at the spatial coordinates $(x_q, y_q, z_l)$, the complex transmission coefficient $t(x_q, y_q, z_l)$ can be represented as a function of its material thickness $h_q^l$. This relationship can be mathematically expressed as:

$$t\left(x_q, y_q, z_l\right) = \exp\left(\frac{-2\pi\kappa h_q^l}{\lambda}\right) \exp\left(\frac{-j2\pi(n - n_{air})h_q^l}{\lambda}\right) \quad (5)$$

Here, $n$ and $\kappa$ correspond to the refractive index and extinction coefficient of the selected dielectric material at $\lambda$, corresponding to the real and imaginary components of the complex refractive index $\tilde{n}$, i.e., $\tilde{n} = n + j\kappa$. For the diffractive OPC designs used for experimental validation, $n$ and $\kappa$ were set based on measurements using a terahertz spectroscopy system[39]. In the other diffractive OPC designs used for numerical analyses at a single wavelength in the terahertz spectrum, the refractive index $n$ was set as 1.7, while $\kappa$ was chosen as 0. For the diffractive multi-wavelength OPC designs shown in Supplementary Fig. S1b, the refractive index profile $n(\lambda)$ was set based on the dispersion of N-BK7 glass[52]. $\kappa(\lambda)$ was set to 0, as the absorption of this material within the visible spectrum is negligible. For each diffractive feature, the thickness value $h$ is composed of two parts: a constant $h_{base}$ that serves as the substrate support and a variable $h_{learnable}$, i.e.,

$$h = h_{learnable} + h_{base} \quad (6)$$

where $h_{learnable}$ represents the learnable thickness value of each diffractive feature and is constrained within the range $[0, h_{max}]$. For the diffractive design shown in Fig. 2b, $h_{max}$ is set as 1.07 mm, covering a full phase modulation range from 0 to $2\pi$ for $\lambda = 0.75$ mm. $h_{base}$ is empirically chosen as 0.2 mm to provide the substrate (mechanical) support for the diffractive features. For the diffractive multi-wavelength OPC design shown in Supplementary Fig. S1b, $h_{max}$ is set

as 1465 nm, ensuring complete phase modulation coverage, ranging from 0 to $2\pi$, for the longest wavelength ($\lambda_{N_w}$). $h_{base}$ is empirically chosen as 200 nm.

We employed the band-limited angular spectrum approach[29,46] to simulate the free-space propagation of coherent optical fields between the diffractive layers, where the resulting field is subsequently modulated by the transmittance $t(x, y, z_{l+1})$ of the $(l+1)^{th}$ diffractive layer. This process can be written as:

$$u_p^{l+1}(x, y, z_{l+1}) = t(x, y, z_{l+1})\mathcal{F}^{-1}\left\{\mathcal{F}\left\{u_q^l(x, y, z_{l+1})\right\}H_q^l\left(f_x, f_y, d_m\right)\right\} \quad (7)$$

where $\mathcal{F}\{\cdot\}$ and $\mathcal{F}^{-1}\{\cdot\}$ denote the 2D fast Fourier transform and the inverse 2D fast Fourier transform operations, respectively, and $H_q^l(f_x, f_y, d_m)$ is the transfer function of free-space propagation with a distance $d_m$ between two successive layers, which is given by:

$$H_q^l\left(f_x, f_y, d\right) = \begin{cases} \exp\left\{\frac{j2\pi d_m}{\lambda}\sqrt{1 - \left(\lambda f_x\right)^2 - \left(\lambda f_y\right)^2}\right\}, & f_x^2 + f_y^2 < \frac{1}{\lambda^2} \\ 0, & f_x^2 + f_y^2 \geq \frac{1}{\lambda^2} \end{cases} \quad (8)$$

where $f_x$ and $f_y$ represent the spatial frequencies along the x and y directions, respectively.

For numerical simulations of the diffractive OPC designs in the terahertz spectrum, the spatial sampling rate of the simulated complex fields is set as 0.4 mm, i.e., ~$0.53\lambda$. The lateral dimension of the individual diffractive features on the diffractive layers was also set as 0.4 mm. As for the numerical simulations of the broadband diffractive designs in the visible spectrum, the spatial sampling rate for simulating the complex fields and the lateral dimension of diffractive features were both selected as 200 nm, i.e., ~$0.35\lambda_m$. The axial spacing between the adjacent layers (including the diffractive layers and input/output planes) was selected as $12\lambda$ for the numerical design shown in Fig. 1a, $20\lambda$ for the experimental validation design shown in Fig. 5a, and $20\lambda_m$ for the multi-wavelength numerical designs shown in Supplementary Fig. S1b.

### Numerical implementation of the diffractive OPC processors and phase-conjugate mirrors

In our diffractive OPC network design, a phase-only object is set to be positioned at $z = z_0$, featuring a phase profile $\Psi(x, y)$ and uniform distribution of unit amplitude. This object is illuminated by a coherent, uniform plane wave, generating an input complex field $i$ that can be mathematically described as:

$$i(x, y) = e^{j\Psi(x, y)} \quad (9)$$

Here $i$ can also be denoted as $u^1$, i.e., $u^1(x, y, z_0) = e^{j\Psi(x, y)}$. Subsequently, as the input light propagates through the diffractive network volume, the input field $i$ (or $u^1$) is subject to a series of diffractive layer modulations and secondary wave formations detailed in the last subsection, ultimately resulting in an output complex field $o(x, y) = u^K(x, y, z_K)$.

For the diffractive OPC processor design depicted in Fig. 2a, both the input and output apertures are designed to have a circular shape with a diameter of ~$59.36\lambda$. The input/output apertures are discretized into 560 pixels, i.e., $N_i = N_o = 560$, with each pixel measuring a size of ~$2.12\lambda \times 2.12\lambda$. To ensure the successful execution of the desired OPC task, each diffractive layer within this diffractive OPC network is designed to contain $200 \times 200$ diffractive features, spanning an area of ~$106\lambda \times 106\lambda$. For the experimental design shown in Fig. 5a, the input and output apertures are square-shaped and share identical dimensions of ~$29.68\lambda \times 29.68\lambda$. The input/output apertures are sampled into arrays of $28 \times 28$ pixels, leading to an individual pixel size of

~ $1.06\lambda \times 1.06\lambda$. The diffractive layers in this design contain $120 \times 120$ diffractive features (for each layer), spanning an area of ~ $64\lambda \times 64\lambda$.

For the diffractive phase-conjugate mirror design illustrated in Fig. 10a, the settings of the input/output apertures and the diffractive layers align with those of the diffractive OPC processor design depicted in Fig. 2a. The main change in this set-up is the integration of a standard mirror placed to the right of the diffractive layers, and the output aperture coincides with the location of the input aperture ($z = z_0$), allowing OPC to operate in reflection mode. In the numerical simulation, a $\pi$ phase shift was introduced at the standard mirror reflection.

For the $N_w = 8$ diffractive multi-wavelength OPC processor shown in Supplementary Fig. S1b, the input and output apertures are designed to have a circular shape with a diameter of ~ $38.92\lambda_m$. The input/output apertures are discretized into 560 pixels, i.e., $N_i = N_o = 560$, with each pixel measuring a size of ~ $1.39\lambda_m \times 1.39\lambda_m$. Each diffractive layer within this diffractive OPC network is designed to contain $646 \times 646$ diffractive features, spanning an area of ~ $225\lambda_m \times 225\lambda_m$. Other multispectral designs analyzed in Supplementary Fig. S2 use a similar architecture but vary by including different numbers of diffractive features within their layers.

## Training loss function and performance metrics

The primary objective of training a diffractive OPC processor is to ensure that its diffractive output field exhibits a phase distribution that is the conjugate of its input counterpart, while concurrently maintaining a uniform amplitude distribution identical to that of the input field. Due to the power loss of the optical field that occurs during its propagation through the diffractive volume, it is necessary to normalize the output field in our training and evaluation process to ensure that the calculated errors and metrics are not influenced by the output diffraction efficiency. In addition, when quantifying the phase conjugation-related errors, it is necessary to eliminate the influence of the phase offset (overall constant phase difference) that might exist between the output phase profile and the ground truth. In this regard, we adopted a normalization strategy[65] for the diffractive output fields and the corresponding ground truth, which are given by:

$$\hat{o} = \sigma o \tag{10}$$

$$\hat{o}^{(GT)} = \sigma^{(GT)} o^{(GT)} \tag{11}$$

where $\sigma^{(GT)}$ and $\sigma$ are the normalization factors, given by:

$$\sigma^{(GT)} = \frac{1}{\sqrt{\sum_{(x,y)\in \mathcal{S}} |o^{(GT)}(x,y)|^2}} \tag{12}$$

$$\sigma = \frac{\sum_{(x,y)\in \mathcal{S}} \sigma^{(GT)} o^{(GT)}(x,y) o^*(x,y)}{\sum_{(x,y)\in \mathcal{S}} |o(x,y)|^2} \tag{13}$$

Based on these normalized quantities constructed above, we devised a custom training loss function $\mathcal{L}$ to achieve a balance between the phase conjugation loss term $\mathcal{L}_{OPC}$ and an amplitude uniformity-related loss term $\mathcal{L}_{Amp}$ (for a single wavelength channel), which can be written as:

$$\mathcal{L} = \mathcal{L}_{OPC} + \beta_{Amp} \mathcal{L}_{Amp} \tag{14}$$

Herein, $\beta_{Amp}$ denotes the weight coefficient associated with $\mathcal{L}_{Amp}$ and is empirically selected as 1 during the training of all the designs presented in this work. The phase conjugation loss term $\mathcal{L}_{OPC}$ penalizes

the MAE between the phase profile of the normalized diffractive output field $\angle\hat{o}$ and its ground truth $\angle\hat{o}^{(GT)}$, which can be written as:

$$\mathcal{L}_{OPC} = \frac{1}{N_{\mathcal{S}}} \sum_{(x,y)\in \mathcal{S}} \left| \angle\hat{o}^{(GT)}(x,y) - \angle\hat{o}(x,y) \right| \tag{15}$$

where $\mathcal{S}$ denotes the output aperture and $N_{\mathcal{S}}$ represents the total number of pixels within $\mathcal{S}$. The amplitude uniformity loss term $\mathcal{L}_{Amp}$ is defined as:

$$\mathcal{L}_{Amp} = \max\left(V, \mathcal{L}_{AmpMSE}\right) - V \tag{16}$$

where $\mathcal{L}_{AmpMSE}$ stands for the normalized mean square error (MSE) between $|\hat{o}|$ and its ground truth $|\hat{o}^{(GT)}|$, which can be written as:

$$\mathcal{L}_{AmpMSE} = \frac{1}{N_{\mathcal{S}}} \sum_{(x,y)\in \mathcal{S}} \left| |\hat{o}^{(GT)}(x,y)| - |\hat{o}(x,y)| \right|^2 \tag{17}$$

In Eq. (16), $V$ is a predetermined threshold used to determine when to start penalizing $\mathcal{L}_{AmpMSE}$. Stated differently, the amplitude uniformity penalty is effective only when $\mathcal{L}_{AmpMSE} > V$. For all the diffractive models in this work, the value of $V$ was empirically chosen as 0.02.

In our experimental validation and our output diffraction efficiency-related analyses, we employed a modified loss function by further adding an output diffraction efficiency-related loss term, $\mathcal{L}_{Eff}$, into the original loss function defined in Eq. (14), which is given by:

$$\mathcal{L} = \mathcal{L}_{OPC} + \beta_{Amp} \mathcal{L}_{Amp} + \beta_{Eff} \mathcal{L}_{Eff} \tag{18}$$

where $\beta_{Eff}$ denotes the weight coefficient associated with $\mathcal{L}_{Eff}$. $\mathcal{L}_{Eff}$ is defined as:

$$\mathcal{L}_{Eff} = e^{-\gamma\eta} \tag{19}$$

where $\gamma$ is an empirical weight coefficient. During the training of the experimental model, the values of $\beta_{Eff}$ and $\gamma$ were set as 1 and 100, respectively; for the diffractive models trained with output diffraction efficiency penalty shown in Fig. 9a, the values of $\gamma$ were selected as 30, 20, 10, 5, and the value of $\beta_{Eff}$ was constantly chosen as 1. $\eta$ represents the output diffraction efficiency and is defined as:

$$\eta = \frac{\sum_{(x,y)\in \mathcal{S}} |o(x,y)|^2}{\sum_{(x,y)\in \mathcal{S}} |i(x,y)|^2} \tag{20}$$

For the optimization of a diffractive multi-wavelength OPC design, we used a loss function that averages the loss values for different wavelength channels computed using Eq. (14) (or Eq. (18) when the power efficiency-related penalty is used). The resulting loss function $\mathcal{L}_{total}$ is given by:

$$\mathcal{L}_{total} = \frac{1}{N_w} \sum_{w=1}^{N_w} \alpha_w \mathcal{L}_w \tag{21}$$

where $\alpha_w$ represents the weight coefficient associated with the loss calculated from the $w^{th}$ wavelength channel of the diffractive multispectral OPC processor. Throughout the training process, the values of $\alpha_w$, all initialized as 1, were dynamically updated after each epoch, guided by the comparative loss magnitudes across the different wavelength channels to achieve a balanced spectral response. This adjustment equation for $\alpha_w$ is expressed as:

$$\alpha_w \leftarrow \max(0.1 \times (\mathcal{L}_w - \mathcal{L}_{mean}) + \alpha_w, 0) \tag{22}$$

where $\mathcal{L}_{\text{mean}}$ represents the mean loss across all the wavelength channels. Under this training approach, a wavelength channel with a loss exceeding the average will see an increase in its $\alpha_w$, thereby raising its balance weight and intensifying the penalty on its output performance.

For evaluating the phase conjugation performance of our presented diffractive OPC processor, we calculated the mean absolute errors of diffractive output amplitude profiles and the diffractive output phase profiles, defined as:

$$MAE_{\text{Amp}} = \frac{1}{N_{\mathcal{S}}} \sum_{(x,y)\in\mathcal{S}} \left| |\hat{o}^{(\text{GT})}(x,y)| - |\hat{o}(x,y)| \right| \tag{23}$$

$$MAE_{\text{Phase}} = \frac{1}{N_{\mathcal{S}}\alpha_{\text{test}}\pi} \sum_{(x,y)\in\mathcal{S}} \left| \angle\hat{o}^{(\text{GT})}(x,y) - \angle\hat{o}(x,y) \right| \tag{24}$$

Here, the testing phase contrast parameter $\alpha_{\text{test}}$ is used to normalize the phase error with regard to the dynamic range of the input phase contrast. Therefore, the error metric $MAE_{\text{Phase}}$ reveals a relative difference between the diffractive output phase profiles and their ground truth.

### Details of the experimental diffractive OPC system

For the experimental set-up, we designed a diffractive OPC processer composed of three isotropic, phase-only diffractive layers ($L_1 - L_3$) between two identical random unknown phase perturbation planes with a phase contrast parameter of $\alpha_{\text{test}} = 0.5$, axially spanning a distance of $80\lambda$ from the first phase perturbation to the second phase perturbation plane, with a distance $d_m$ of $20\lambda$ between successive diffractive layers. At the system's forefront, we placed a square-shaped aperture with a width of ~$4.3\lambda$, serving as a small pinpoint object $i$. This initiates an input wavefront similar to a spherical wave, which travels a distance ($d$) of ~$266\lambda$ and subsequently encounters a random, unknown phase perturbation plane with $28 \times 28$ pixels. The illustrative design is shown in Fig. 5a.

To optimize the diffractive OPC network shown in Fig. 5b, we created a training dataset comprised of 50,000 randomly generated phase perturbation planes. Each of these planes was conceptualized as a phase-only mask, with complex transmission coefficients, $t_P(x,y)$, defined as:

$$t_P(x,y) = \exp\left( j\frac{2\pi\Delta n}{\lambda}P(x,y) \right) \tag{25}$$

The random height map $P(x,y)$ is defined as:

$$P(x,y) = V(x,y) * K(\sigma) \tag{26}$$

where $V(x,y)$ follows a normal distribution with a mean $\mu$ and a standard deviation $\sigma_0$, i.e.,

$$V(x,y) \sim N(\mu, \sigma_0^2) \tag{27}$$

$K(\sigma)$ represents a Gaussian smoothing kernel with zero mean and a standard deviation of $\sigma$. The symbol "*" stands for the 2D convolution operation. Throughout this work, we used $\mu = 25\lambda$, $\sigma_0 = 4\lambda$, and $\sigma = 8\lambda$ for the generation of the 50,000 random phase perturbation planes. Given these settings, the resulting average correlation length of phase perturbation can be calculated as $L \sim 14.7\lambda$ using a phase auto-correlation function[36,54].

As shown in Fig. 5c, a terahertz continuous wave (CW) system was used to test our diffractive OPC design. In this system, we used a terahertz source consisting of a modular amplifier/multiplier chain (AMC) (Virginia Diode Inc. WR9.0 M SGX/WR4.3x2 WR2.2x2), coupled with a compatible diagonal horn antenna (Virginia Diode Inc. WR2.2). A 10-dBm radiofrequency (RF) input signal was generated at a frequency of 11.1111 GHz (fRF1) at the input of the AMC and was then multiplied by 36 times to generate the output CW radiation at 0.4 THz, corresponding to a wavelength of 0.75 mm. Also, the AMC output was modulated with a 1- kHz square wave for lock-in detection. The 4- mm-width input aperture was positioned ~50 mm away from the exit aperture of the horn antenna. The intensity distribution within the output aperture was 2D-scanned at a step size of 0.8 mm by a single-pixel mixer (Virginia Diode Inc. WRI 2.2), which was mounted on an XY positioning stage constructed using two Thorlabs NRT100 motorized stages. A 10-dBm RF signal at 11.0833 GHz (fRF2) was also received by the detector to serve as the local oscillator to down-convert the output frequency to 1 GHz. This down-converted signal was then amplified using a low-noise amplifier (Mini-Circuits ZRL-1150-LN + ) with a gain of 80 dBm and filtered through a band-pass filter at 1 GHz ( +/- 10 MHz) (KL Electronics 3C40-1000/T10-O/O), which mitigate the noise resulted from unwanted frequency bands. After the signal went through a tunable attenuator (HP 8495B) for linear calibration, it was then processed by a low-noise power detector (Mini-Circuits ZX47-60). The resulting output voltage from the detector was measured using a lock-in amplifier (Stanford Research SR830), where the 1- kHz square wave was used as the reference signal for calibration into a linear scale. The same system was also used to test our diffractive multi-wavelength OPC processor design, where the CW radiation was set to operate at the wavelengths of 0.75 mm, 0.775 mm, and 0.8 mm.

For the fabrication of the resulting diffractive OPC processor, a 3D printer (Objet30 Pro, Stratasys) was used to fabricate the diffractive layers shown in Figs. 5b and 7a. The phase perturbation planes and the input aperture were also 3D printed using the same printer (Objet30 Pro, Stratasys). To achieve alignment in accordance with our optical forward model of the experimental diffractive design, a 3D-printed holder was utilized to assemble the input aperture, the phase perturbation planes and the printed diffractive layers, ensuring their precise 3D positioning.

### Training and numerical implementation details

The numerical modeling and training of the diffractive OPC designs presented in this work were implemented using Python (v3.7.13) and PyTorch (version 1.12.1, Meta Platform Inc.). We selected the Adam optimizer with the default parameters in PyTorch for optimizing the trainable parameters, i.e., the phase modulation coefficients of the diffractive layers. For all the models presented in this paper, the training underwent 100 epochs, with the batch size set as 128. Starting from an initial value of 0.001, the learning rate was set to decay at a rate of 0.5 every 10 epochs. These diffractive models were trained using a workstation with an Nvidia GeForce RTX 3090 GPU, an Intel Core i9-11900 CPU, and 128 GB of RAM. The training of an 8-layer diffractive OPC design typically took ~20 hours.

### Reporting summary

Further information on research design is available in the Nature Portfolio Reporting Summary linked to this article.

## Data availability

All the data and methods needed to evaluate the conclusions of this work are presented in the main text and Supporting Information. Additional data can be requested from the corresponding author.

## Code availability

The codes used in this work use standard libraries and scripts that are publicly available in PyTorch.

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

## Acknowledgements
Ozcan Research Lab at UCLA acknowledges the funding of the U.S. Department of Energy (DOE), Office of Basic Energy Sciences, Division of Materials Sciences and Engineering under Award # DE-SC0023088. Jarrahi Lab acknowledges the support of the NSF PFI program.

## Author contributions
A.O. conceived and initiated the research. C.S., J.L., and T.G. conducted the experiments and processed the resulting data. C.S. and J.L. conducted numerical simulations. C.S., J.L., and Y.L. contributed to the code implementation. All the authors contributed to the preparation of the manuscript. A.O. and M.J. supervised the research.

## Competing interests
A.O., J.L., and C.S. are co-inventors of a pending patent application on the presented method. The remaining authors declare no competing interests.
