## [Peer Review File · Nature Communications]

All-Optical Phase Conjugation Using Diffractive Wavefront ProcessingREVIEWER COMMENTS

Reviewer #1 (Remarks to the Author):

The manuscript by Shen et.al shows another contribution from the Ozcan group in the series of optical processors. This time, the optical processor, designed with multiple engineered diffractive layers, produces the optical phase conjugation wave of the optical wave input. The significance of work is that the design, data for training and validation, were all done in computer with precise forward model of optical propagation; then demonstrations were done in the lab with fabricated diffractive layers in accordance with the design.

On one hand, the results are attractive and encouraging for the community. On the other hand, after several demonstrations from the group for different optical processors but similar approach along this line (Ref. 29,31, 32, 34, 39, 40, 41), we expect some tangible demonstrations to showcase the real-world applications of the techniques in order to publish in Nat. Comms. The current demonstrations with proof-of-concept are very much expected while the approach is not new. In my opinion, the current form is worth to publish in a more specialized journal.

From the technical point of view, it is helpful if the authors describe the reasons for all the choices of parameters used in the simulation such as 0.4mm (0.53λ), or diameter of 59.36λ , etc.

For propagation, the light will expand. For instance, if we monitor the wavefront at the far distance (or infinity), the wavefront will be super large, but the digital simulation will have the same number of pixels for both input and output planes. Therefore, in the simulation the sampling rate will decrease at the later layers. But it is fixed at 0.4mm in the current work. Hope that the authors can clarify this.

Reviewer #2 (Remarks to the Author):

Reviewer's reports on the manuscript: All-Optical Phase Conjugation Using Diffractive Wavefront Processing by Che-Yung Shen et al.

Authors present the design and experimental validation of a diffractive wavefront processor to approximate all-optical phase conjugation operation for input fields with phase aberrations. By adopting a deep-learning strategy it was possible to achieve this by using passive passive diffractive layers. Arbitrary phase-aberrated wavefront was used from an input aperture and thus producing a conjugate wavefront of the input wave. The proof was obtained by fabricating 3D diffractive layers. Training using deep learning allowed ot perform performing optical phase conjugation (OPC) on phase distortions that were never seen by the diffractive processor during its training. THz radiation, was adopted for the diffractive processor. Moreover, authors showed transmissive as well as diffractive phase-conjugation mirror.

Authors claim compactness and scalable nature for their processor. One main advantage lies in the passive nature of the system, OPC concept is a very important topic with applications for many purposes, such as turbidity suppression or aberration correction. Applications span from biomedical to aerospace where wavefront analysis or engineering is necessary for focusing as well as imaging. Such tool could be really of great importance!

The theoretical formulation appears to be correct. The content is very clear, and results are amazing. In fact, the possibility to have a tool capable to perform phase conjugation of any complex wavefields avoiding digital computation, image acquisition and/or or active wavefront modulation is of huge importance. They demonstrate the operability for many case of aberrations/distortions.

Some comment/questions:

The main question is if this processor it is scalable to shorter wavelengths and in case how??

Tolerance of the distance and angle misalignments of the different layers;
What about the spatial resolution? What can happen by increasing the spatial resolution (or reduction of pixel size) what about in this case for "misalignment" problems?
Is it implementable by multiplexing for different wavelengths/bands

About references

Maybe the reader can benefit to have in the list of refs. A recent review on strategies for removing wavefront aberrations; so the suggestion is to add the following paper:

Sirico, Daniele Gaetano, et al. "Compensation of aberrations in holographic microscopes: main strategies and applications." *Applied Physics B* 128.4 (2022): 78.

One more query is about the possibility to manipulate wavefront with Zernike 3D descriptor. Such processor could be of some help about 3D case?? Please look at this:

Loss Minimized Data Reduction in Single-Cell Tomographic Phase Microscopy Using 3D Zernike Descriptors. *Intell Comput.* 2023;2:0010.DOI:10.34133/icomputing.0010

In summary the paper is very good and it deserves to be published.

Authors should provide more discussions on perspective to scaling down the system for shorter wavelengths.

Provided that authors revise by answering the above comments it can be published.

Reviewer #3 (Remarks to the Author):

The manuscript, entitled "All-optical phase conjugation using diffractive wavefront processing" by Shen et al, has realized a new function, i.e., optical phase conjugation (OPC), via a well-designed diffractive deep neural network, which transfers the digital computation into the real-world terahertz applications. Regardless of the technical realizations, the novelty is not sufficient to support its publications in *Nature Communications*, since many similar works in the same framework have been presented. More specific comments are listed below:

1. Overall, this work is quite similar to the other studies published by the same group, especially a very recent published NC paper [*Nat. Commun.* 14(1):6830, 2023], which has exactly the same experimental setup as this study but merely a varied design of the diffractive neural network, with different functionalities. That means, such framework regarding the diffractive layers in terahertz regime has been well demonstrated and reported. Therefore, I question the novelty about this study since it is indeed an incremental trial.

2. The authors demonstrated that with diffractive layers serve as a phase conjugate mirror, AOPC is realized. Nevertheless, it seems that the proposed method can only deal with simple optical pattern (combination of three weighted Zernike polynomials at most), probably due to the limited generalization of the learning-based implementations (e.g., the network is fed with Zernike samples) and defects occur in the fabrication. It therefore brings limited interest, without the demonstrations for more universal/arbitrary optical patterns, like random phase profile, speckle pattern or other complicated patterns.

3. About the power efficiency: although the diffraction efficiency is directly related to the power efficiency, the damage threshold for the diffractive layers should be discussed since it may determine the available and potential applications.

4. About the misalignment: looking at the phase profiles of the diffractive layers (Fig. 2b and Fig. 9b), all the layers should be aligned concentrically. How does the misalignment affect the OPC performance?

5. About the OPC speed: I do agree that previous OPC implementations are subjected to low speed (down to ms order these days) due to their inevitable separated steps (i.e., writing and reading).

The proposed method essentially combines these two steps in a single diffractive network. Though the authors have claimed the proposed OPC operates in an ultra-high-speed manner, what dynamic or fast-changing scenarios can be addressed by the proposed method is out of presentation.

6. About the deep learning part: in the training loss, MAE and MSE are used to guide the convergence of network for phase and amplitude, respectively, which is believed to be the proper choices to refine the modulation in the authors' practice. Nevertheless, the presented results still show inefficient amplitude modulation. Is it due to the insufficient training or is it one of the drawbacks in the terahertz realizations?

7. Following the last question, in the reflected realizations, the layers of networks are equivalently doubled but weaker amplitude modulation can be seen (Fig. 9), compared to the transmitted mode. So, does the accumulated attenuation or fabrication error among those diffractive layers limit or even weaken its performance?

We sincerely thank the referees for their reviews and the constructive feedback that we have received on our manuscript “**All-Optical Phase Conjugation Using Diffractive Wavefront Processing**”.

As detailed below, we have revised our manuscript in response to the reviewers’ comments. The changes that we have made in the manuscript files have been highlighted in yellow. We briefly summarize below some of the major changes that we have made to our manuscript.

Summary of our Revisions:

- We have demonstrated multi-wavelength diffractive OPC processors capable of achieving all-optical phase conjugation simultaneously across a broad spectrum of wavelengths, where at each wavelength channel, the aberrated input wavefronts were independent of the other wavelength channels, representing a highly challenging broadband OPC task. Our numerical analyses utilized wavelengths within the visible spectrum, confirming the capability of these diffractive processor models to successfully perform multi-wavelength phase conjugation, regardless of the material dispersion used.
- We have also conducted additional experiments to validate the effectiveness of the multi-wavelength diffractive OPC processor concept. This involved the creation of a diffractive OPC processor designed to operate at three distinct wavelengths within the terahertz spectrum. The outcomes of this experiment demonstrated the diffractive processor’s ability to simultaneously perform phase conjugation at multiple wavelengths, with an independent aberration in each channel.
- Addressing the reviewers’ concerns regarding misalignment and performance robustness, we have broadened our analyses to encompass the effects of random misalignments. Moreover, we have illustrated the feasibility of implementing a "vaccination" strategy to significantly improve the diffractive OPC processors’ performance in the face of random, unpredictable misalignments between the diffractive layers and the input/output apertures.
- We have also enhanced our manuscript by including analyses related to the complexity of the input phase aberration patterns. Our findings clearly demonstrate that our diffractive processor designs maintain consistent OPC performance across a wide range of phase aberrations (including the combinations of many randomly selected Zernike polynomials), highlighting its extensive generalization.
- We have further clarified that by incorporating a diffraction efficiency penalty in the training loss function and enhancing the system’s degrees of freedom through an increased number of diffractive features, our method can achieve an even higher power efficiency (as demonstrated in **Figures 8-9**), also broadening the scope of operation beyond the limits of traditional OPC methods.

The following items have been revised or newly added, all highlighted in yellow in our manuscript files:

Revised sections:

- Abstract
- Introduction
- Results
- Discussion
- Methods
- Supplementary Information

Renumbered Figures:

Previous	New
Fig. 7	Fig. 8

Fig. 8	Fig. 9
Fig. 9	Fig. 10
Supplementary Fig. S1	Supplementary Fig. S5
Supplementary Fig. S2	Supplementary Fig. S6

New Sub-sections Added:

- New sub-section added to the **Results**:
 - **Optical phase conjugation of multi-wavelength illumination**

Revised Figures (all the figure #s below refer to the #s after renumbering):

- **Fig. 10**: The diffractive phase-conjugate mirror design and the visualization of the output field examples.

New Figures Added:

- **Fig. 7**: Experimental results of the transmissive diffractive multi-wavelength OPC processor design.
 - **Supplementary Fig. S1**: Schematic and operation mechanism of a transmissive diffractive multi-wavelength OPC processor.
 - **Supplementary Fig. S2**: Spectral multiplexing capacity and the scalability analysis of diffractive multi-wavelength OPC processors.
 - **Supplementary Fig. S3**: Visualization of the multi-wavelength OPC processor output fields using a broadband diffractive processor operating at the visible band.
 - **Supplementary Fig. S4**: Visualization of the multi-wavelength OPC output fields using a broadband diffractive processor employing a dispersion-free material.
 - **Supplementary Fig. S7**: Phase profiles of the resulting layers for the diffractive phase-conjugate mirror design.
 - **Supplementary Fig. S8**: Output visualization of the comparison between the transmissive OPC processor designs and the reflective OPC processor designs using different amplitude MSE thresholds (V) used during the training stage.
 - **Supplementary Fig. S9**: Results for testing the external generalization performance of the diffractive OPC processor design.
 - **Supplementary Fig. S10**: The impact of lateral misalignments on the phase conjugation performance of the vaccinated diffractive OPC processors.
 - **Supplementary Fig. S11**: Output visualization of vaccinated diffractive OPC processors with different degrees of lateral misalignments.
 - **Supplementary Fig. S12**: The impact of axial misalignments on the phase conjugation performance of the vaccinated diffractive OPC processors.
 - **Supplementary Fig. S13**: Output visualization of vaccinated diffractive OPC processors with different degrees of axial misalignments.
-

Reviewer #1 (Remarks to the Author):

The manuscript by Shen et.al shows another contribution from the Ozcan group in the series of optical processors. This time, the optical processor, designed with multiple engineered diffractive layers, produces the optical phase conjugation wave of the optical wave input. The significance of work is that the design, data for training and validation, were all done in computer with precise forward model of optical propagation; then demonstrations were done in the lab with fabricated diffractive layers in accordance with the design.

On one hand, the results are attractive and encouraging for the community.

-- We sincerely thank the reviewer for his/her constructive evaluations.

1. On the other hand, after several demonstrations from the group for different optical processors but similar approach along this line (Ref. 29, 31, 32, 34, 39, 40, 41), we expect some tangible demonstrations to showcase the real-world applications of the techniques in order to publish in Nat. Comms. The current demonstrations with proof-of-concept are very much expected while the approach is not new. In my opinion, the current form is worth to publish in a more specialized journal.

-- We sincerely appreciate the reviewer's insights. In terms of novelty and the major advantages of our diffractive OPC processor presented in this work compared to the existing literature, we can list the following:

(1) In this work, the necessity for precise control over the output complex field **presents a significant challenge in diffractive optical information processing, approximating a nonlinear OPC function that has not been explored yet in the literature.** To address this challenge, we created a new optimization process to minimize a specially devised loss function (detailed in Methods), which compares both the normalized amplitude and phase differences between the diffractive output fields o and their corresponding ground truth $o^{(GT)}$. More details regarding the training loss function, numerical forward model and structural parameters of the diffractive OPC processor can be found in the Methods section.

(2) Our study showcases **the first demonstration** of a passive broadband OPC system capable of phase conjugation across multiple distinct wavelengths, eliminating wavelength-specific limitations. This pivotal capability significantly differs from previous methodologies in OPC and creates a solid alternative to wavefront shaping and aberration compensation solutions across a large set of distinct wavelengths, where at each wavelength channel, the aberrated input wavefronts are independent of the other wavelength channels, representing a highly challenging broadband OPC task. To substantiate the effectiveness of our approach, we have designed a multi-wavelength OPC system, conducted numerical simulations, and experimentally validated our framework. See the newly added **Figure 7** as well as **Supplementary Figs. S1, S2, S3 and S4**, together with a new subsection titled **“Optical phase conjugation of multi-wavelength illumination”**, added to the Results section of our revised manuscript. Quoted below:

“Optical phase conjugation of multi-wavelength illumination:

*Our diffractive OPC processor model previously presented in **Fig. 2** was designed solely for operation at a single wavelength and was numerically demonstrated at the terahertz part of the spectrum. In the following numerical analyses, we present designs of diffractive OPC processors that are capable of multispectral operation and demonstrate their efficacy within the visible spectrum (400-750 nm), where at each wavelength channel, the aberrated input wavefronts are independent of the other wavelength channels, representing a challenging broadband OPC task. As illustrated in **Supplementary Fig. S1a**, these multi-wavelength diffractive designs simultaneously perform phase conjugation of aberrated input*

wavefronts at N_w distinct wavelengths $\{\lambda_1, \lambda_2, \dots, \lambda_{N_w}\}$, uniformly distributed within a range of 400 nm to 750 nm. In these broadband diffractive OPC processor designs, we used $N_w = 2, 4, \text{ and } 8$. Given this wavelength multiplexing task at hand, the total number of trainable diffractive features (N) in the OPC processor was scaled proportionally with the number of wavelength channels (N_w) to maintain the information processing capability per channel. To analyze this behavior for each N_w choice, we created different OPC processors with $N = \{0.5N_iN_oN_w, N_iN_oN_w, 2N_iN_oN_w\}$ and $K = 12$. Here, N_i and N_o represent the number of diffraction-limited pixels within the input and output apertures, respectively. In these numerical analyses, we selected N-BK7 glass as the diffractive material due to its prevalent use for optical components at the visible band⁶⁴. The training data and methods used for these visible band diffractive OPC processor designs follow their terahertz counterparts. After the training, the diffractive layer thickness profiles corresponding to the model with $N = 2N_iN_oN_w$ and $N_w = 8$ wavelengths are visualized in **Supplementary Fig. S1b** as an example.

To evaluate the performance of these diffractive multi-wavelength OPC processors, we performed numerical blind testing using input fields with phase profiles composed of 2 random Zernike polynomials, which randomly varied at different wavelengths and were never used in the training. The average phase errors for all these diffractive OPC models with different N and N_w values are summarized on the left part of **Supplementary Fig. S2**, revealing an error level of $<1.5\%$ for all the diffractive models. These results also indicate that introducing more degrees of freedom in the design (by increasing N) can substantially improve the phase accuracy of multispectral OPC operation; for example, for all the cases of $N_w = 2, 4 \text{ and } 8$, the average phase error is reported as $\sim 1.4\%$ when using $N = 0.5N_iN_oN_w$, which reduced to $\sim 1.0\%$ when using $N = 2N_iN_oN_w$. Exemplary multi-wavelength output fields from the diffractive OPC model using $N = 2N_iN_oN_w$ and $N_w = 8$ wavelengths are provided in **Supplementary Fig. S3a**, demonstrating a decent match with their ground truth fields across all the eight wavelength channels. We also tested the same diffractive OPC model with structurally more complex input fields composed of 3 random Zernike polynomials (never used in the training); as shown in **Supplementary Fig. S3b**, the diffractive processor still revealed output phase profiles that closely align with their ground truth across all the 8 wavelength channels. All these blind testing results corroborate our framework's feasibility for performing multi-wavelength OPC operations at different parts of the spectrum.

These multi-wavelength diffractive OPC designs shown on the left part of **Supplementary Fig. S2** employed N-BK7 as the diffractive layer material, which has a known dispersion curve with varying refractive index values as a function of the wavelength, which was numerically modeled in our diffractive designs. Next, we investigated the feasibility of constructing a broadband diffractive OPC processor using a material with a different dispersion property; to highlight an extreme case, we assumed a diffractive material with flat dispersion such that the refractive index does not change as a function of the wavelength within the operation band of interest. We selected this dispersion-free material to highlight an important feature of multispectral diffractive OPC designs: their phase conjugation performance at different wavelengths is **independent** of the material dispersion. Therefore, in this analysis, we adopted a diffractive material with flat dispersion, exhibiting a constant refractive index ($n=1.7$) within its operational band (400 nm to 750 nm). As reported on the right part of **Supplementary Fig. S2**, these dispersion-free OPC diffractive designs achieve average phase errors of 1.34% and 0.95% across $N_w = 8$ wavelength channels when using $N = 0.5N_iN_oN_w$ and $2N_iN_oN_w$, respectively. Remarkably, they attain performance parity with previous designs that utilized N-BK7, even exhibiting a slight improvement. The output examples of the dispersion-free OPC processor design using $N = 2N_iN_oN_w$ and $N_w = 8$ wavelengths are also provided in **Supplementary Fig. S4**, all presenting very good agreement with the ground truth. These findings confirm that the broadband OPC capability of our diffractive processors is not subject to specific dispersion properties of the materials, thereby highlighting it as a versatile platform that can adapt to varying material choices and operation wavelengths.”

Additionally, we have included new experimental results of a diffractive multi-wavelength OPC system operating in the terahertz spectrum. The related content has been added to the Results section of our revised manuscript, together with the newly added **Figure 7**, quoted below:

*“... Apart from the monochromatic diffractive OPC design reported above, we also performed experimental validation of a multi-wavelength OPC processor. Following the same architecture and training method used by the monochromatic model shown in **Fig. 5**, we trained a diffractive multi-wavelength OPC processor to simultaneously support phase conjugation operation at three distinct wavelengths: 0.75, 0.775 and 0.8 mm. The resulting diffractive layers were fabricated using 3D printing, as shown in **Fig. 7a**. In this experiment, we employed the same input aperture as in the previous test and sequentially illuminated the phase perturbation plane with each operational wavelength, capturing the corresponding intensity distribution within the output aperture. The experimental results, presented in **Fig. 7b**, show that our diffractive multi-wavelength OPC processor consistently generated Gaussian-shaped circular output patterns for all three wavelengths, achieving similar results to its monochromatic counterpart shown in **Fig. 6b**. This success further substantiates the feasibility of designing diffractive OPC processors capable of correcting broadband aberrated fields across multiple wavelength channels, with an independent aberration in each channel...”*

New text related to the experimental demonstration of our diffractive multi-wavelength OPC system has also been added to the revised Introduction section:

“... Beyond this single-wavelength diffractive OPC design, we also trained and fabricated a diffractive multi-wavelength OPC design and demonstrated its successful operation at three distinct wavelengths (0.75 mm, 0.775 mm and 0.8 mm), experimentally validating the feasibility of our broadband OPC framework in generating multi-wavelength phase-conjugated fields...”

(3) The presented diffractive OPC platform holds design flexibility since it can function at different parts of the electromagnetic spectrum without retraining or optimizing its diffractive layers by simply scaling its features in proportion to the illumination wavelength. This feature makes our diffractive OPC framework highly desirable for correcting wavefront distortions at various parts of the spectrum where **cost-effective and easy-to-implement phase conjugation solutions do not readily exist, such as the IR and THz bands**. Moreover, since isotropic dielectric materials are used to fabricate these diffractive layers, the **OPC function is independent of the polarization state of the incident light**, which remains unchanged at the output of the diffractive OPC system. Combining these unique advantages, our diffractive OPC framework holds promise for various applications, including, e.g., turbidity suppression and phase aberration correction, and can unlock different opportunities across diverse areas, including but not limited to biomedical imaging, microscopy, telescope systems and optical communication.

(4) Our diffractive OPC processor enables **wavefront reshaping at the speed of light, along with high power efficiency (see Figures 8 and 9 of the main text)**— a capability not widely available in previous nonlinear crystal-based analog OPC systems. Some of these advantages of our diffractive OPC processors have been added in our revised Discussion section, which is quoted below:

“...Another key benefit of our framework is its exceptionally fast response time, performing OPC at the speed of light propagation, a feature paramount for numerous wavefront shaping (WFS) applications. An optimal WFS technique for scattering suppression would need to combine rapid system response, power efficient generation of the conjugated wave, and extensive degrees of freedom for precise wave manipulation^{15,61}. Speed is especially crucial for WFS, as it must be accomplished within the speckle correlation period, typically <1 ms, to be compatible with the dynamic nature of various specimens, such as living organisms and tissue samples. This requirement poses a challenge for conventional DOPC methods due to inherent delays in their operation. In contrast, the presented diffractive OPC processor, being entirely optical, achieves OPC operation as the light is transmitted through a thin

*optical volume, thus effortlessly satisfying the crucial demand for rapid response. By incorporating a diffraction efficiency penalty in the training loss function and enhancing the system's degrees of freedom through an increased number of diffractive features, our method holds the potential of achieving even higher power efficiency (see e.g., **Figs. 8-9**), also broadening the scope of operation beyond the limits of traditional OPC methods. These advantages render our framework particularly well suited for applications in biomedical imaging and potentially for astronomical observations through scattering media such as the atmosphere, where energy efficiency is critically important⁶²...*

Finally, the novelty and the unique advantages of our diffractive multi-wavelength OPC approach have been elaborated on in the revised Discussion section, which is quoted below:

*"...The presented diffractive OPC processor designs offer significant advantages with their multi-wavelength operation, distinctly setting them apart from conventional OPC methods. While conventional AOPC systems are typically limited to narrowband operation due to their dependency on wavelength-specific nonlinear coefficients of materials, our diffractive OPC processors employ basic dielectric materials without the need for a specific dispersion relationship (see **Supplementary Fig. S2**). Our diffractive designs allow for multi-wavelength OPC operations, facilitating efficient performance over a wider spectral range, independent from the dispersion characteristics of the diffractive materials. For example, we showcased a broadband diffractive OPC processor designed using N-BK7, a widely used borosilicate glass known for its high damage threshold and superior optical quality, making it well-suited for a vast spectrum of wavelengths in the visible. In applications that demand a higher damage threshold, fused silica emerges as a viable alternative for the diffractive layer material, with UV to near-infrared band transmission and negligible thermal expansion⁶⁰..."*

2. From the technical point of view, it is helpful if the authors describe the reasons for all the choices of parameters used in the simulation such as 0.4mm (0.53 λ), or diameter of 59.36 λ , etc. For propagation, the light will expand. For instance, if we monitor the wavefront at the far distance (or infinity), the wavefront will be super large, but the digital simulation will have the same number of pixels for both input and output planes. Therefore, in the simulation the sampling rate will decrease at the later layers. But it is fixed at 0.4mm in the current work. Hope that the authors can clarify this.

-- We appreciate the reviewer's inquiry regarding the parameter choices in our simulations, particularly concerning the decision to maintain a fixed sampling rate despite the natural expansion of the light wavefront during propagation. To address this, we have elaborated on our rationale and the methodology used by incorporating additional explanations into the revised Results section, quoted below:

"...It is worth noting that, in the analyses presented in this paper, we demonstrated OPC operations with a unit magnification between the input and output apertures, axially separated by several tens of wavelengths. This design choice ensures a relatively high diffraction efficiency within the output aperture of the OPC processor, and also allows the precise modeling of the free space propagation between adjacent diffractive planes using the angular spectrum method⁴⁷, with a lateral sampling of $\sim\lambda/2$ at the input and the propagated fields. If phase conjugation with magnification or demagnification at the output field is desired, the implementation of a pyramidal architecture for the diffractive OPC processor⁴⁸ may be considered. In that case, the free-space propagation can be modeled using a scalable angular spectrum method⁴⁹ or the Fresnel diffraction approach to enable different sampling sizes at the input and output fields. These approaches can allow for the efficient design and modeling of diffractive OPC processors, accommodating a wider range of structural parameters and applications..."

Reviewer #2 (Remarks to the Author):

Authors present the design and experimental validation of a diffractive wavefront processor to approximate all-optical phase conjugation operation for input fields with phase aberrations. By adopting a deep-learning strategy it was possible to achieve this by using passive passive diffractive layers. Arbitrary phase-aberrated wavefront was used from an input aperture and thus producing a conjugate wavefront of the input wave. The proof was obtained by fabricating 3D diffractive layers. Training using deep learning allowed to perform optical phase conjugation (OPC) on phase distortions that were never seen by the diffractive processor during its training. THz radiation, was adopted for the diffractive processor. Moreover, authors showed transmissive as well as diffractive phase-conjugation mirror.

Authors claim compactness and scalable nature for their processor. One main advantage lies in the passive nature of the system, OPC concept is a very important topic with applications for many purposes, such as turbidity suppression or aberration correction. Applications span from biomedical to aerospace where wavefront analysis or engineering is necessary for focusing as well as imaging. Such tool could be really of great importance!

The theoretical formulation appears to be correct. The content is very clear, and results are amazing. In fact, the possibility to have a tool capable to perform phase conjugation of any complex wavefields avoiding digital computation, image acquisition and/or or active wavefront modulation is of huge importance. They demonstrate the operability for many case of aberrations/distortions.

-- We sincerely thank the reviewer for the constructive and positive evaluation.

Some comment/questions:

1. The main question is if this processor it is scalable to shorter wavelengths and in case how??

-- In response to the reviewer's query regarding scalability to shorter wavelengths, we have expanded our research to include a new diffractive OPC design aimed at broadband operation at the visible range. New **Supplementary Figures S1, S2, S3 and S4** have been added to showcase the related results and analyses, together with a new subsection of "**Optical phase conjugation of multi-wavelength illumination**", added into the Results section of our revised manuscript:

*"...Our diffractive OPC processor model previously presented in **Fig. 2** was designed solely for operation at a single wavelength and was numerically demonstrated at the terahertz part of the spectrum. In the following numerical analyses, we present designs of diffractive OPC processors that are capable of multispectral operation and demonstrate their efficacy within the visible spectrum (400-750 nm), where at each wavelength channel, the aberrated input wavefronts are independent of the other wavelength channels, representing a challenging broadband OPC task. As illustrated in **Supplementary Fig. S1a**, these multi-wavelength diffractive designs simultaneously perform phase conjugation of aberrated input wavefronts at N_w distinct wavelengths $\{\lambda_1, \lambda_2, \dots, \lambda_{N_w}\}$, uniformly distributed within a range of 400 nm to 750 nm. In these broadband diffractive OPC processor designs, we used $N_w = 2, 4, \text{ and } 8$. Given this wavelength multiplexing task at hand, the total number of trainable diffractive features (N) in the OPC processor was scaled proportionally with the number of wavelength channels (N_w) to maintain the information processing capability per channel. To analyze this behavior for each N_w choice, we created different OPC processors with $N = \{0.5N_iN_oN_w, N_iN_oN_w, 2N_iN_oN_w\}$ and $K = 12$. Here, N_i and N_o represent the number of diffraction-limited pixels within the input and output apertures, respectively. In these numerical analyses, we selected N-BK7 glass as the diffractive material due to its prevalent use for optical components at the visible band⁵⁴. The training data and methods used for these visible band diffractive OPC processor designs follow their terahertz*

counterparts. After the training, the diffractive layer thickness profiles corresponding to the model with $N = 2N_iN_oN_w$ and $N_w = 8$ wavelengths are visualized in **Supplementary Fig. S1b** as an example. To evaluate the performance of these diffractive multi-wavelength OPC processors, we performed numerical blind testing using input fields with phase profiles composed of 2 random Zernike polynomials, which randomly varied at different wavelengths and were never used in the training. The average phase errors for all these diffractive OPC models with different N and N_w values are summarized on the left part of **Supplementary Fig. S2**, revealing an error level of $<1.5\%$ for all the diffractive models. These results also indicate that introducing more degrees of freedom in the design (by increasing N) can substantially improve the phase accuracy of multispectral OPC operation; for example, for all the cases of $N_w = 2, 4$ and 8 , the average phase error is reported as $\sim 1.4\%$ when using $N = 0.5N_iN_oN_w$, which reduced to $\sim 1.0\%$ when using $N = 2N_iN_oN_w$. Exemplary multi-wavelength output fields from the diffractive OPC model using $N = 2N_iN_oN_w$ and $N_w = 8$ wavelengths are provided in **Supplementary Fig. S3a**, demonstrating a decent match with their ground truth fields across all the eight wavelength channels. We also tested the same diffractive OPC model with structurally more complex input fields composed of 3 random Zernike polynomials (never used in the training); as shown in **Supplementary Fig. S3b**, the diffractive processor still revealed output phase profiles that closely align with their ground truth across all the 8 wavelength channels. All these blind testing results corroborate our framework's feasibility for performing multi-wavelength OPC operations at different parts of the spectrum.

These multi-wavelength diffractive OPC designs shown on the left part of **Supplementary Fig. S2** employed N-BK7 as the diffractive layer material, which has a known dispersion curve with varying refractive index values as a function of the wavelength, which was numerically modeled in our diffractive designs. Next, we investigated the feasibility of constructing a broadband diffractive OPC processor using a material with a different dispersion property; to highlight an extreme case, we assumed a diffractive material with flat dispersion such that the refractive index does not change as a function of the wavelength within the operation band of interest. We selected this dispersion-free material to highlight an important feature of multispectral diffractive OPC designs: their phase conjugation performance at different wavelengths is **independent** of the material dispersion. Therefore, in this analysis, we adopted a diffractive material with flat dispersion, exhibiting a constant refractive index ($n=1.7$) within its operational band (400 nm to 750 nm). As reported on the right part of **Supplementary Fig. S2**, these dispersion-free OPC diffractive designs achieve average phase errors of 1.34% and 0.95% across $N_w = 8$ wavelength channels when using $N = 0.5N_iN_oN_w$ and $2N_iN_oN_w$, respectively. Remarkably, they attain performance parity with previous designs that utilized N-BK7, even exhibiting a slight improvement. The output examples of the dispersion-free OPC processor design using $N = 2N_iN_oN_w$ and $N_w = 8$ wavelengths are also provided in **Supplementary Fig. S4**, all presenting very good agreement with the ground truth. These findings confirm that the broadband OPC capability of our diffractive processors is not subject to specific dispersion properties of the materials, thereby highlighting it as a versatile platform that can adapt to varying material choices and operation wavelengths..."

We have also added new text related to the diffractive multi-wavelength OPC processor for broadband light in the Introduction and Methods sections:

(In the Introduction section)

"... Additionally, we demonstrated our diffractive processor's capability for multi-wavelength OPC operation, showcasing its practicality for applications spanning different spectral bands."

(In the Methods section)

"...For the $N_w = 8$ diffractive multi-wavelength OPC processor shown in **Supplementary Fig. S1b**, the

input and output apertures are designed to have a circular shape with a diameter of $\sim 38.92\lambda_m$. The input/output apertures are discretized into 560 pixels, i.e., $N_i = N_o = 560$, with each pixel measuring a size of $\sim 1.39\lambda_m \times 1.39\lambda_m$. Each diffractive layer within this diffractive OPC network is designed to contain 646×646 diffractive features, spanning an area of $\sim 225\lambda_m \times 225\lambda_m$. Other multispectral designs analyzed in **Supplementary Fig. S2** use a similar framework but vary by including different numbers of diffractive features within their layers...

“...For the optimization of a diffractive multi-wavelength OPC design, we used a loss function that averages the loss values for different wavelength channels computed using Eq. (14) (or Eq. (18) when the power efficiency-related penalty is used). The resulting loss function \mathcal{L}_{total} is given by:

$$\mathcal{L}_{total} = \frac{1}{N_w} \sum_{w=1}^{N_w} \alpha_w \mathcal{L}_w \quad (21),$$

where α_w represents the weight coefficient associated with the loss calculated from the w^{th} wavelength channel of the diffractive multispectral OPC processor. Throughout the training process, the values of α_w , initialized as 1, were dynamically updated after each epoch, guided by the comparative loss magnitudes across the different wavelength channels to achieve a relatively balanced spectral performance. This adjustment equation for α_w is expressed as:

$$\alpha_w \leftarrow \max(0.1 \times (\mathcal{L}_w - \mathcal{L}_{mean}) + \alpha_w, 0) \quad (22),$$

where \mathcal{L}_{mean} represents the mean loss across all wavelength channels. Under this training approach, a wavelength channel with a loss exceeding the average will see an increase in its α_w , thereby raising its balance weight and intensifying the penalty on its output performance...”

2. Tolerance of the distance and angle misalignments of the different layers;

-- We thank the reviewer for pointing out a critical aspect regarding the practical implementation of diffractive OPC processors. Mechanical misalignments between the diffractive layers could indeed affect the OPC performance of our diffractive processors. We have evaluated the performance of our diffractive processors under different degrees of misalignments and added new contents in the Discussion section of our manuscript, together with new **Supplementary Figs. S10, S11, S12 and S13**, quoted below:

“... In the practical implementations of our diffractive OPC processors, mechanical misalignment between different elements could constitute a notable challenge to their phase conjugation performance, as they can cause the optical waves to be modulated by the diffractive layers in an undesired way, leading to results deviating from their designed performance. To provide further evidence for this, we utilized the diffractive OPC processor model previously shown in **Fig. 2a** and subjected the diffractive layers to different magnitudes of random displacements, either in the lateral directions ($\Delta_x, \Delta_y \in U[-\Delta_{xy,test}, \Delta_{xy,test}]$) or the axial direction ($\Delta_z \in U[-\Delta_{z,tr}, \Delta_{z,tr}]$), sampled from random uniform distributions (U). The resulting OPC performances of these misaligned diffractive processors are summarized in **Supplementary Figs. S10 and S12** using blue curves, which reveal a clear trend of increasing degradation in both the amplitude and phase errors as the misalignment gets more severe. ...”

To mitigate this issue, we have introduced a “**vaccination**” strategy during the training process. The efficacy of our vaccination strategy has been demonstrated and discussed in the Discussion section of our revised manuscript, together with **Supplementary Figs. S10, S11, S12 and S13**, as quoted below:

“... To address this misalignment challenge, a “**vaccination**” strategy can be applied during the training process by modeling these misalignment errors as random noise into the numerical forward model of

the system^{34,37,57}. Specifically, the 3D random displacements of the diffractive layers (Δ_x , Δ_y and Δ_z) can be modeled using random variables, changing from iteration to iteration during the training process, to provide substantial resilience against such random displacements at a tolerable cost of performance loss. The efficacy of this vaccination-based design strategy is demonstrated in **Supplementary Figs. S10 and S12**, where new diffractive OPC processor models were trained under random lateral and axial misalignments of varying magnitudes. These vaccinated diffractive OPC processor results (shown in orange and green curves) reveal that if the training parameters $\Delta_{xy,tr}$ and $\Delta_{z,tr}$ encompass the range of misalignments encountered in the blind testing ($\Delta_{xy,test}$ and $\Delta_{z,test}$), the impact of such physical/random misalignments on OPC performance can be significantly reduced. For instance, compared to the baseline model that was not “vaccinated” against these misalignments, the average output phase error of the “vaccinated” model trained using $\Delta_{xy,tr} = 0.24\lambda$ is reduced to 3.30% when tested with $\Delta_{xy,test} = 0.24\lambda$. Visual analyses reported in **Supplementary Fig. S11 and S13** further corroborate these findings, showing that the diffractive processor maintains high OPC output accuracy under different levels of misalignments, even when using structurally more complex aberrated input fields that are formed by a greater number of Zernike polynomials, as illustrated in **Supplementary Figs. S11b and S13b**. These results highlight our diffractive OPC processor’s capability to accommodate various phase aberration patterns and withstand unexpected random 3D misalignments among the diffractive layers. It is also possible to adopt the same vaccination strategy to mitigate the impact of rotational misalignments between the diffractive layers when such misalignments become a critical factor in experiments. Specifically, in-plane rotations of the diffractive layers can be modeled with 2D coordinate transformations based on unitary rotation matrices, and the out-of-plane rotations can be accounted for by adjusting the wave propagation forward model between randomly tilted diffractive planes^{57–59}. ...”

3. What about the spatial resolution? What can happen by increasing the spatial resolution (or reduction of pixel size) what about in this case for “misalignment” problems?

-- To address this question of the referee about the spatial resolution, we conducted additional blinded tests of our diffractive OPC processors using input phase fields constituted by a greater number of randomly selected Zernike polynomials, which represent structurally more complex phase profiles and exhibit higher spatial resolution. These analyses are reported in the newly added **Supplementary Figs. S9, S11 and S13**, as well as in the Discussion section of our revised manuscript, quoted as follows:

“...To better understand our diffractive OPC models’ generalization capability, we deliberately increased the complexity of the test data by generating input phase profiles with ≥ 3 Zernike polynomials, introducing phase structures that are spatially more complex, exhibiting rapid variations. This blind testing evaluation was conducted using the same diffractive phase-conjugate mirror model that was originally trained on phase profiles constructed from two randomly selected Zernike polynomials. The results of this evaluation, demonstrated in **Supplementary Fig. S9a**, reveal that our diffractive OPC processor maintained high accuracy levels, with phase errors remaining $<1.3\%$ across input datasets incorporating 3 to 8 randomly selected Zernike polynomials. Furthermore, the examples of the output fields, depicted in **Supplementary Figs. S9b-d**, showcase very good concordance with their corresponding targets. These results highlight the OPC processor’s versatility and effectiveness, affirming its capability to accurately handle a broad range of wavefront patterns, including various highly structured phase distortions and aberrations (never seen before). This underscores the utility of the presented platform as a general-purpose optical phase conjugation device...”

“... Visual analyses reported in **Supplementary Fig. S11 and S13** further corroborate these findings, showing that the diffractive processor maintains high OPC output accuracy under different levels of misalignments, even when using structurally more complex aberrated input fields that are formed by a greater number of Zernike polynomials, as illustrated in **Supplementary Figs. S11b and S13b**. These

results highlight our diffractive OPC processor's capability to accommodate various phase aberration patterns and withstand unexpected random 3D misalignments among the diffractive layers. ...”

4. Is it implementable by multiplexing for different wavelengths/bands

Can train multispectral design

-- Following these valuable suggestions of the reviewer, we have designed passive broadband OPC diffractive processors capable of phase conjugation across multiple distinct wavelengths, eliminating wavelength-specific limitations. This pivotal capability significantly differs from previous methodologies in OPC and creates a solid alternative to wavefront shaping and aberration compensation solutions across a large set of distinct wavelengths, where at each wavelength channel, the aberrated input wavefronts are independent of the other wavelength channels, representing a highly challenging broadband OPC task. To substantiate the effectiveness of our approach, we have designed a multi-wavelength OPC system, conducted numerical simulations, and experimentally validated our framework. See the newly added **Figure 7** as well as **Supplementary Figs. S1, S2, S3 and S4**, together with a new subsection titled **“Optical phase conjugation of multi-wavelength illumination”**, added to the Results section of our revised manuscript. Quoted below:

“Optical phase conjugation of multi-wavelength illumination:

*Our diffractive OPC processor model previously presented in **Fig. 2** was designed solely for operation at a single wavelength and was numerically demonstrated at the terahertz part of the spectrum. In the following numerical analyses, we present designs of diffractive OPC processors that are capable of multispectral operation and demonstrate their efficacy within the visible spectrum (400-750 nm), where at each wavelength channel, the aberrated input wavefronts are independent of the other wavelength channels, representing a challenging broadband OPC task. As illustrated in **Supplementary Fig. S1a**, these multi-wavelength diffractive designs simultaneously perform phase conjugation of aberrated input wavefronts at N_w distinct wavelengths $\{\lambda_1, \lambda_2, \dots, \lambda_{N_w}\}$, uniformly distributed within a range of 400 nm to 750 nm. In these broadband diffractive OPC processor designs, we used $N_w = 2, 4, \text{ and } 8$. Given this wavelength multiplexing task at hand, the total number of trainable diffractive features (N) in the OPC processor was scaled proportionally with the number of wavelength channels (N_w) to maintain the information processing capability per channel. To analyze this behavior for each N_w choice, we created different OPC processors with $N = \{0.5N_iN_oN_w, N_iN_oN_w, 2N_iN_oN_w\}$ and $K = 12$. Here, N_i and N_o represent the number of diffraction-limited pixels within the input and output apertures, respectively. In these numerical analyses, we selected N-BK7 glass as the diffractive material due to its prevalent use for optical components at the visible band⁶⁴. The training data and methods used for these visible band diffractive OPC processor designs follow their terahertz counterparts. After the training, the diffractive layer thickness profiles corresponding to the model with $N = 2N_iN_oN_w$ and $N_w = 8$ wavelengths are visualized in **Supplementary Fig. S1b** as an example.*

*To evaluate the performance of these diffractive multi-wavelength OPC processors, we performed numerical blind testing using input fields with phase profiles composed of 2 random Zernike polynomials, which randomly varied at different wavelengths and were never used in the training. The average phase errors for all these diffractive OPC models with different N and N_w values are summarized on the left part of **Supplementary Fig. S2**, revealing an error level of $<1.5\%$ for all the diffractive models. These results also indicate that introducing more degrees of freedom in the design (by increasing N) can substantially improve the phase accuracy of multispectral OPC operation; for example, for all the cases of $N_w = 2, 4$ and 8 , the average phase error is reported as $\sim 1.4\%$ when using $N = 0.5N_iN_oN_w$, which reduced to $\sim 1.0\%$ when using $N = 2N_iN_oN_w$. Exemplary multi-wavelength output fields from the diffractive OPC model using $N = 2N_iN_oN_w$ and $N_w = 8$ wavelengths are provided in **Supplementary Fig. S3a**, demonstrating a decent match with their ground truth fields across all the eight wavelength channels. We also tested the same diffractive OPC model with structurally more*

complex input fields composed of 3 random Zernike polynomials (never used in the training); as shown in **Supplementary Fig. S3b**, the diffractive processor still revealed output phase profiles that closely align with their ground truth across all the 8 wavelength channels. All these blind testing results corroborate our framework's feasibility for performing multi-wavelength OPC operations at different parts of the spectrum.

These multi-wavelength diffractive OPC designs shown on the left part of **Supplementary Fig. S2** employed N-BK7 as the diffractive layer material, which has a known dispersion curve with varying refractive index values as a function of the wavelength, which was numerically modeled in our diffractive designs. Next, we investigated the feasibility of constructing a broadband diffractive OPC processor using a material with a different dispersion property; to highlight an extreme case, we assumed a diffractive material with flat dispersion such that the refractive index does not change as a function of the wavelength within the operation band of interest. We selected this dispersion-free material to highlight an important feature of multispectral diffractive OPC designs: their phase conjugation performance at different wavelengths is **independent** of the material dispersion. Therefore, in this analysis, we adopted a diffractive material with flat dispersion, exhibiting a constant refractive index ($n=1.7$) within its operational band (400 nm to 750 nm). As reported on the right part of **Supplementary Fig. S2**, these dispersion-free OPC diffractive designs achieve average phase errors of 1.34% and 0.95% across $N_w = 8$ wavelength channels when using $N = 0.5N_iN_oN_w$ and $2N_iN_oN_w$, respectively. Remarkably, they attain performance parity with previous designs that utilized N-BK7, even exhibiting a slight improvement. The output examples of the dispersion-free OPC processor design using $N = 2N_iN_oN_w$ and $N_w = 8$ wavelengths are also provided in **Supplementary Fig. S4**, all presenting very good agreement with the ground truth. These findings confirm that the broadband OPC capability of our diffractive processors is not subject to specific dispersion properties of the materials, thereby highlighting it as a versatile platform that can adapt to varying material choices and operation wavelengths.”

Additionally, we have included new experimental results of a diffractive multi-wavelength OPC system operating in the terahertz spectrum. The related content has been added to the Results section of our revised manuscript, together with the newly added **Figure 7**, quoted below:

“...Apart from the monochromatic diffractive OPC design reported above, we also performed experimental validation of a multi-wavelength OPC processor. Following the same architecture and training method used by the monochromatic model shown in **Fig. 5**, we trained a diffractive multi-wavelength OPC processor to simultaneously support phase conjugation operation at three distinct wavelengths: 0.75, 0.775 and 0.8 mm. The resulting diffractive layers were fabricated using 3D printing, as shown in **Fig. 7a**. In this experiment, we employed the same input aperture as in the previous test and sequentially illuminated the phase perturbation plane with each operational wavelength, capturing the corresponding intensity distribution within the output aperture. The experimental results, presented in **Fig. 7b**, show that our diffractive multi-wavelength OPC processor consistently generated Gaussian-shaped circular output patterns for all three wavelengths, achieving similar results to its monochromatic counterpart shown in **Fig. 6b**. This success further substantiates the feasibility of designing diffractive OPC processors capable of correcting broadband aberrated fields across multiple wavelength channels, with an independent aberration in each channel...”

New text related to the experimental demonstration of a diffractive multi-wavelength OPC system has also been added to the revised Introduction section:

“... Beyond this single-wavelength diffractive OPC design, we also trained and fabricated a diffractive multi-wavelength OPC design and demonstrated its successful operation at three distinct wavelengths (0.75 mm, 0.775 mm and 0.8 mm), experimentally validating the feasibility of our broadband OPC framework in generating multi-wavelength phase-conjugated fields...”

5. About references

Maybe the reader can benefit of have in the list of refs. A recent review on strategies for removing wavefront aberrations; so the suggestion is to add the following paper: Sirico, Daniele Gaetano, et al. "Compensation of aberrations in holographic microscopes: main strategies and applications." *Applied Physics B* 128.4 (2022): 78.

-- We thank the reviewer for providing the important literature relevant to the contents of our manuscript. This reference has been cited as Ref. 21 in the following sentence, Introduction section: "...*This unique technique later fostered myriad applications including, but not limited to, laser beam focusing through scattering media⁵⁻¹⁶, imaging through turbid materials¹⁷⁻²⁰, compensation of aberration in microscopy²¹, and improving the performance of optical communication systems²²⁻²⁶, among others...*"

6. One more query is about the possibility to manipulate wavefront with Zernike 3D descriptor. Such processor could be of some help about 3D case?? Please look at this: Loss Minimized Data Reduction in Single-Cell Tomographic Phase Microscopy Using 3D Zernike Descriptors. *Intell Comput.* 2023;2:0010.DOI:10.34133 /icomputing.0010

-- Regarding the inquiry about the potential for wavefront manipulation using Zernike 3D descriptors, we appreciate the reviewer's valuable insights. While the focus of our current manuscript does not extend to 3D imaging technologies, this valuable suggestion indeed aligns with some research efforts within our team. Specifically, we have been examining the capabilities of a wavelength-multiplexed diffractive optical processor for 3D/tomographic quantitative phase imaging. This research could indeed benefit from and relate closely to the concept of Zernike 3D descriptors, which are left as future work.

In summary the paper is very good and it deserves to be published. Authors should provide more discussions on perspective to scaling down the system for shorter wavelegths.

Provided that authors revise by answering the above comments it can be published.

Reviewer #3 (Remarks to the Author):

The manuscript, entitled "All-optical phase conjugation using diffractive wavefront processing" by Shen et al, has realized a new function, i.e., optical phase conjugation (OPC), via a well-designed diffractive deep neural work, which transfers the digital computation into the real-world terahertz applications. Regardless of the technical realizations, the novelty is not sufficient to support its publications in *Nature Communications*, since many similar works in the same framework have been presented. More specific comments are listed below:

1. Overall, this work is quite similar to the other studies published by the same group, especially a very recent published NC paper [*Nat. Commun.* 14(1):6830, 2023], which has exactly the same experimental setup as this study but merely a varied design of the diffractive neural network, with different functionalities. That means, such framework regarding the diffractive layers in terahertz regime has been well demonstrated and reported. Therefore, I question the novelty about this study since it is indeed an incremental trial.

-- We thank the referee for their valuable and thoughtful comments.

First, we would like to kindly clarify an important confusion that we observe. The referee's comment regarding our experimental set-up being identical to an earlier paper of our team appears to stem from a confusion: the THz source and detector shown in our previous paper are the same source and detector used in our experiments, but the entire diffractive OPC processor concept, its design, optimization and fabrication as well as experimental validation are unique to this work.

Perhaps an analogy could be useful here: **Let's assume we have an expensive laser or a microscope in our lab. That piece of equipment will surely appear in many publications as it is used to measure different quantities in different papers. This is exactly the same in our experimental figures: we simply used the same source and detector, which are not novel. The novelty here include the concept, design, optimization, operation/analyses and experimental validation of our diffractive OPC processors.**

For our esteemed referee, we are sure this confusion will be cleared by summarizing that a major equipment of a light source or measurement device will naturally appear in different papers and this does not take away anything from the novelty of our paper.

Furthermore, the method and the diffractive device demonstrated in our study are uniquely distinct from our previous work [Nat. Commun. 14(1):6830, 2023] in several critical aspects. Firstly, unlike the hybrid approach highlighted in the previous work that integrates an *electronic neural network encoder* with an optical diffractive decoder, our approach in this paper is all-optical, focusing on diffractive OPC via wavefront engineering. Secondly, while our previous work aimed at transferring optical information through an opaque occlusion – **essentially a zero-transmittance amplitude mask where the optical phase is undefined**, our research in this work is dedicated to mitigating phase perturbations utilizing the time-reversal properties of OPC, thereby producing a conjugated version of the input aperture's distorted phase at the output. Lastly, this work achieved precise control over the output complex field, and it presents a significant challenge in diffractive optical information processing, **all-optically approximating a nonlinear OPC function that has not been explored yet.**

In terms of novelty and the major advantages of our diffractive OPC processor presented in this work compared to the existing literature, we can list the following:

(1) In this work, the necessity for precise control over the output complex field **presents a significant challenge in diffractive optical information processing, approximating a nonlinear OPC function that has not been explored yet in the literature.** To address this challenge, we created a new optimization process to minimize a specially devised loss function (detailed in Methods), which compares both the normalized amplitude and phase differences between the diffractive output fields o and their corresponding ground truth $o^{(GT)}$. More details regarding the training loss function, numerical forward model and structural parameters of the diffractive OPC processor can be found in the Methods section.

(2) Our study showcases **the first demonstration** of a passive broadband OPC system capable of phase conjugation across multiple distinct wavelengths, eliminating wavelength-specific limitations. This pivotal capability significantly differs from previous methodologies in OPC and creates a solid alternative to wavefront shaping and aberration compensation solutions across a large set of distinct wavelengths, where at each wavelength channel, the aberrated input wavefronts are independent of the other wavelength channels, representing a highly challenging broadband OPC task. To substantiate the effectiveness of our approach, we have designed a multi-wavelength OPC system, conducted numerical simulations, and experimentally validated our framework. See the newly added **Figure 7** as well as **Supplementary Figs. S1, S2, S3 and S4**, together with a new subsection titled **“Optical phase conjugation of multi-wavelength illumination”**, added to the Results section of our revised manuscript. Quoted below:

“Optical phase conjugation of multi-wavelength illumination:

Our diffractive OPC processor model previously presented in Fig. 2 was designed solely for operation at a single wavelength and was numerically demonstrated at the terahertz part of the spectrum. In the following numerical analyses, we present designs of diffractive OPC processors that are capable of multispectral operation and demonstrate their efficacy within the visible spectrum (400-750 nm), where at each wavelength channel, the aberrated input wavefronts are independent of the other wavelength

channels, representing a challenging broadband OPC task. As illustrated in **Supplementary Fig. S1a**, these multi-wavelength diffractive designs simultaneously perform phase conjugation of aberrated input wavefronts at N_w distinct wavelengths $\{\lambda_1, \lambda_2, \dots, \lambda_{N_w}\}$, uniformly distributed within a range of 400 nm to 750 nm. In these broadband diffractive OPC processor designs, we used $N_w = 2, 4, \text{ and } 8$. Given this wavelength multiplexing task at hand, the total number of trainable diffractive features (N) in the OPC processor was scaled proportionally with the number of wavelength channels (N_w) to maintain the information processing capability per channel. To analyze this behavior for each N_w choice, we created different OPC processors with $N = \{0.5N_iN_oN_w, N_iN_oN_w, 2N_iN_oN_w\}$ and $K = 12$. Here, N_i and N_o represent the number of diffraction-limited pixels within the input and output apertures, respectively. In these numerical analyses, we selected N-BK7 glass as the diffractive material due to its prevalent use for optical components at the visible band⁵⁴. The training data and methods used for these visible band diffractive OPC processor designs follow their terahertz counterparts. After the training, the diffractive layer thickness profiles corresponding to the model with $N = 2N_iN_oN_w$ and $N_w = 8$ wavelengths are visualized in **Supplementary Fig. S1b** as an example.

To evaluate the performance of these diffractive multi-wavelength OPC processors, we performed numerical blind testing using input fields with phase profiles composed of 2 random Zernike polynomials, which randomly varied at different wavelengths and were never used in the training. The average phase errors for all these diffractive OPC models with different N and N_w values are summarized on the left part of **Supplementary Fig. S2**, revealing an error level of $<1.5\%$ for all the diffractive models. These results also indicate that introducing more degrees of freedom in the design (by increasing N) can substantially improve the phase accuracy of multispectral OPC operation; for example, for all the cases of $N_w = 2, 4 \text{ and } 8$, the average phase error is reported as $\sim 1.4\%$ when using $N = 0.5N_iN_oN_w$, which reduced to $\sim 1.0\%$ when using $N = 2N_iN_oN_w$. Exemplary multi-wavelength output fields from the diffractive OPC model using $N = 2N_iN_oN_w$ and $N_w = 8$ wavelengths are provided in **Supplementary Fig. S3a**, demonstrating a decent match with their ground truth fields across all the eight wavelength channels. We also tested the same diffractive OPC model with structurally more complex input fields composed of 3 random Zernike polynomials (never used in the training); as shown in **Supplementary Fig. S3b**, the diffractive processor still revealed output phase profiles that closely align with their ground truth across all the 8 wavelength channels. All these blind testing results corroborate our framework's feasibility for performing multi-wavelength OPC operations at different parts of the spectrum.

These multi-wavelength diffractive OPC designs shown on the left part of **Supplementary Fig. S2** employed N-BK7 as the diffractive layer material, which has a known dispersion curve with varying refractive index values as a function of the wavelength, which was numerically modeled in our diffractive designs. Next, we investigated the feasibility of constructing a broadband diffractive OPC processor using a material with a different dispersion property; to highlight an extreme case, we assumed a diffractive material with flat dispersion such that the refractive index does not change as a function of the wavelength within the operation band of interest. We selected this dispersion-free material to highlight an important feature of multispectral diffractive OPC designs: their phase conjugation performance at different wavelengths is **independent** of the material dispersion. Therefore, in this analysis, we adopted a diffractive material with flat dispersion, exhibiting a constant refractive index ($n=1.7$) within its operational band (400 nm to 750 nm). As reported on the right part of **Supplementary Fig. S2**, these dispersion-free OPC diffractive designs achieve average phase errors of 1.34% and 0.95% across $N_w = 8$ wavelength channels when using $N = 0.5N_iN_oN_w$ and $2N_iN_oN_w$, respectively. Remarkably, they attain performance parity with previous designs that utilized N-BK7, even exhibiting a slight improvement. The output examples of the dispersion-free OPC processor design using $N = 2N_iN_oN_w$ and $N_w = 8$ wavelengths are also provided in **Supplementary Fig. S4**, all presenting very good agreement with the ground truth. These findings confirm that the broadband OPC capability of our diffractive processors is not subject to specific dispersion properties of the materials, thereby

highlighting it as a versatile platform that can adapt to varying material choices and operation wavelengths.”

Additionally, we have included new experimental results of a diffractive multi-wavelength OPC system operating in the terahertz spectrum. The related content has been added to the Results section of our revised manuscript, together with the newly added **Figure 7**, quoted below:

“... Apart from the monochromatic diffractive OPC design reported above, we also performed experimental validation of a multi-wavelength OPC processor. Following the same architecture and training method used by the monochromatic model shown in **Fig. 5**, we trained a diffractive multi-wavelength OPC processor to simultaneously support phase conjugation operation at three distinct wavelengths: 0.75, 0.775 and 0.8 mm. The resulting diffractive layers were fabricated using 3D printing, as shown in **Fig. 7a**. In this experiment, we employed the same input aperture as in the previous test and sequentially illuminated the phase perturbation plane with each operational wavelength, capturing the corresponding intensity distribution within the output aperture. The experimental results, presented in **Fig. 7b**, show that our diffractive multi-wavelength OPC processor consistently generated Gaussian-shaped circular output patterns for all three wavelengths, achieving similar results to its monochromatic counterpart shown in **Fig. 6b**. This success further substantiates the feasibility of designing diffractive OPC processors capable of correcting broadband aberrated fields across multiple wavelength channels, with an independent aberration in each channel...”

New text related to the experimental demonstration of our diffractive multi-wavelength OPC system has also been added to the revised Introduction section:

“... Beyond this single-wavelength diffractive OPC design, we also trained and fabricated a diffractive multi-wavelength OPC design and demonstrated its successful operation at three distinct wavelengths (0.75 mm, 0.775 mm and 0.8 mm), experimentally validating the feasibility of our broadband OPC framework in generating multi-wavelength phase-conjugated fields...”

(3) The presented diffractive OPC platform holds design flexibility since it can function at different parts of the electromagnetic spectrum without retraining or optimizing its diffractive layers by simply scaling its features in proportion to the illumination wavelength. This feature makes our diffractive OPC framework highly desirable for correcting wavefront distortions at various parts of the spectrum where **cost-effective and easy-to-implement phase conjugation solutions do not readily exist, such as the IR and THz bands**. Moreover, since isotropic dielectric materials are used to fabricate these diffractive layers, the **OPC function is independent of the polarization state of the incident light**, which remains unchanged at the output of the diffractive OPC system. Combining these unique advantages, our diffractive OPC framework holds promise for various applications, including, e.g., turbidity suppression and phase aberration correction, and can unlock different opportunities across diverse areas, including but not limited to biomedical imaging, microscopy, telescope systems and optical communication.

(4) Our diffractive OPC processor enables **wavefront reshaping at the speed of light, along with high power efficiency (see Figures 8 and 9 of the main text)**— a capability not widely available in previous nonlinear crystal-based analog OPC systems. Some of these advantages of our diffractive OPC processors have been added in our revised Discussion section, which is quoted below:

“...Another key benefit of our framework is its exceptionally fast response time, performing OPC at the speed of light propagation, a feature paramount for numerous wavefront shaping (WFS) applications. An optimal WFS technique for scattering suppression would need to combine rapid system response, power efficient generation of the conjugated wave, and extensive degrees of freedom for precise wave manipulation^{15,61}. Speed is especially crucial for WFS, as it must be accomplished within the speckle correlation period, typically <1 ms, to be compatible with the dynamic nature of various specimens, such as living organisms and tissue samples. This requirement poses a challenge for conventional

*DOPC methods due to inherent delays in their operation. In contrast, the presented diffractive OPC processor, being entirely optical, achieves OPC operation as the light is transmitted through a thin optical volume, thus effortlessly satisfying the crucial demand for rapid response. By incorporating a diffraction efficiency penalty in the training loss function and enhancing the system's degrees of freedom through an increased number of diffractive features, our method holds the potential of achieving even higher power efficiency (see e.g., **Figs. 8-9**), also broadening the scope of operation beyond the limits of traditional OPC methods. These advantages render our framework particularly well suited for applications in biomedical imaging and potentially for astronomical observations through scattering media such as the atmosphere, where energy efficiency is critically important⁶²...*

Finally, the novelty and the unique advantages of our diffractive multi-wavelength OPC approach have been elaborated on in the revised Discussion section, which is quoted below:

*"...The presented diffractive OPC processor designs offer significant advantages with their multi-wavelength operation, distinctly setting them apart from conventional OPC methods. While conventional AOPC systems are typically limited to narrowband operation due to their dependency on wavelength-specific nonlinear coefficients of materials, our diffractive OPC processors employ basic dielectric materials without the need for a specific dispersion relationship (see **Supplementary Fig. S2**). Our diffractive designs allow for multi-wavelength OPC operations, facilitating efficient performance over a wider spectral range, independent from the dispersion characteristics of the diffractive materials. For example, we showcased a broadband diffractive OPC processor designed using N-BK7, a widely used borosilicate glass known for its high damage threshold and superior optical quality, making it well-suited for a vast spectrum of wavelengths in the visible. In applications that demand a higher damage threshold, fused silica emerges as a viable alternative for the diffractive layer material, with UV to near-infrared band transmission and negligible thermal expansion⁶⁰..."*

2. The authors demonstrated that with diffractive layers serves as a phase conjugate mirror, AOPC is realized. Nevertheless, it seems that the proposed method can only deal with simple optical pattern (combination of three weighted Zernike polynomials at most), probably due to the limited generalization of the learning-based implementations (e.g., the network is fed with Zernike samples) and defeats occur in the fabrication. It therefore brings limited interest, without the demonstrations for more universal/arbitrary optical patterns, like random phase profile, speckle pattern or other complicated patterns.

-- To address the concerns of generality or generalization of our approach, we conducted additional tests of our diffractive OPC processors using input phase fields constituted by a greater number of Zernike polynomials, which represent structurally much more complex phase profiles exhibiting random phase perturbations never seen during the training stage. We have created a new **Supplementary Fig. S9** to summarize our findings and added additional sentences to the Discussion section of our revised manuscript, quoted as follows:

*"...To better understand our diffractive OPC models' generalization capability, we deliberately increased the complexity of the test data by generating input phase profiles with ≥ 3 Zernike polynomials, introducing phase structures that are spatially more complex, exhibiting rapid variations. This blind testing evaluation was conducted using the same diffractive phase-conjugate mirror model that was originally trained on phase profiles constructed from two randomly selected Zernike polynomials. **The results of this evaluation, demonstrated in Supplementary Fig. S9a, reveal that our diffractive OPC processor maintained high accuracy levels, with phase errors remaining <1.3% across input datasets incorporating 3 to 8 randomly selected Zernike polynomials. Furthermore, the examples of the output fields, depicted in Supplementary Figs. S9b-d, showcase very good concordance with their corresponding targets.***

These results highlight the OPC processor's versatility and effectiveness, affirming its capability to accurately handle a broad range of wavefront patterns, including various highly structured phase distortions and aberrations (never seen before). This underscores the utility of the presented platform as a general-purpose optical phase conjugation device...

3. About the power efficiency: although the diffraction efficiency is directly related to the power efficiency, the damage threshold for the diffractive layers should be discussed since it may determine the available and potential applications.

-- We sincerely thank the reviewer for giving us a chance to further discuss the practical realizations of our diffractive OPC processor. To address these concerns regarding the damage threshold of diffractive layer materials, we have added the following text in the Discussion section, quoted as follows:

*"... The presented diffractive OPC processor designs offer significant advantages with their multi-wavelength operation, distinctly setting them apart from conventional OPC methods. While conventional AOPC systems are typically limited to narrowband operation due to their dependency on wavelength-specific nonlinear coefficients of materials, our diffractive OPC processors employ basic dielectric materials without the need for a specific dispersion relationship (see **Supplementary Fig. S2**). Our diffractive designs allow for multi-wavelength OPC operations, facilitating efficient performance over a wider spectral range, independent from the dispersion characteristics of the diffractive materials. **For example, we showcased a broadband diffractive OPC processor designed using N-BK7, a widely used borosilicate glass known for its high damage threshold and superior optical quality, making it well-suited for a vast spectrum of wavelengths in the visible. In applications that demand a higher damage threshold, fused silica emerges as a viable alternative for the diffractive layer material, with UV to near-infrared band transmission and negligible thermal expansion**⁶⁰."*

4. About the misalignment: looking at the phase profiles of the diffractive layers (Fig. 2b and Fig. 9b), all the layers should be aligned concentrically. How does the misalignment affect the OPC performance?

-- We thank the reviewer for pointing out a critical aspect for the practical implementation of a diffractive processor. Mechanical misalignments between the diffractive layers could indeed affect the OPC performance of our diffractive processor. We have evaluated the performance of our diffractive processor under different degrees of misalignments and added new contents in the Discussion section of our manuscript, together with new **Supplementary Figs. S10, S11, S12 and S13**, quoted below:

*"... In the practical implementations of our diffractive OPC processors, mechanical misalignment between different elements could constitute a notable challenge to their phase conjugation performance, as they can cause the optical waves to be modulated by the diffractive layers in an undesired way, leading to results deviating from their designed performance. To provide further evidence for this, we utilized the diffractive OPC processor model previously shown in **Fig. 2a** and subjected the diffractive layers to different magnitudes of random displacements, either in the lateral directions ($\Delta_x, \Delta_y \in U[-\Delta_{xy,test}, \Delta_{xy,test}]$) or the axial direction ($\Delta_z \in U[-\Delta_{z,tr}, \Delta_{z,tr}]$), sampled from random uniform distributions (U). The resulting OPC performances of these misaligned diffractive processors are summarized in **Supplementary Figs. S10 and S12** using blue curves, which reveal a clear trend of increasing degradation in both the amplitude and phase errors as the misalignment gets more severe. ..."*

To mitigate this issue, we have introduced a "**vaccination**" strategy during the training process. The efficacy of our vaccination strategy has been demonstrated and discussed in the Discussion section of

our revised manuscript, together with **Supplementary Figs. S10, S11, S12 and S13**, as quoted below:

“... To address this misalignment challenge, a “vaccination” strategy can be applied during the training process by modeling these misalignment errors as random noise into the numerical forward model of the system^{34,37,57}. Specifically, the 3D random displacements of the diffractive layers (Δ_x , Δ_y and Δ_z) can be modeled using random variables, changing from iteration to iteration during the training process, to provide substantial resilience against such random displacements at a tolerable cost of performance loss. The efficacy of this vaccination-based design strategy is demonstrated in **Supplementary Figs. S10 and S12**, where new diffractive OPC processor models were trained under random lateral and axial misalignments of varying magnitudes. These vaccinated diffractive OPC processor results (shown in orange and green curves) reveal that if the training parameters $\Delta_{xy,tr}$ and $\Delta_{z,tr}$ encompass the range of misalignments encountered in the blind testing ($\Delta_{xy,test}$ and $\Delta_{z,test}$), the impact of such physical/random misalignments on OPC performance can be significantly reduced. For instance, compared to the baseline model that was not “vaccinated” against these misalignments, the average output phase error of the “vaccinated” model trained using $\Delta_{xy,tr} = 0.24\lambda$ is reduced to 3.30% when tested with $\Delta_{xy,test} = 0.24\lambda$. **Visual analyses reported in Supplementary Fig. S11 and S13 further corroborate these findings, showing that the diffractive processor maintains high OPC output accuracy under different levels of misalignments, even when using structurally more complex aberrated input fields that are formed by a greater number of Zernike polynomials, as illustrated in Supplementary Figs. S11b and S13b. These results highlight our diffractive OPC processor’s capability to accommodate various phase aberration patterns and withstand unexpected random 3D misalignments among the diffractive layers.** It is also possible to adopt the same vaccination strategy to mitigate the impact of rotational misalignments between the diffractive layers when such misalignments become a critical factor in experiments. Specifically, in-plane rotations of the diffractive layers can be modeled with 2D coordinate transformations based on unitary rotation matrices, and the out-of-plane rotations can be accounted for by adjusting the wave propagation forward model between randomly tilted diffractive planes^{57–59}. ...”

5. About the OPC speed: I do agree that previous OPC implementations are subjected to low speed (down to ms order these days) due to their inevitable separated steps (i.e., writing and reading). The proposed method essentially combines these two steps in a single diffractive network. Though the authors have claimed the proposed OPC operates in an ultra-high-speed manner, what dynamic or fast-changing scenarios can be addressed by the proposed method is out of presentation.

-- We thank the reviewer for giving us a chance to highlight our diffractive OPC processor’s advantages in terms of the response speed of OPC operation. The importance of speed in OPC and the corresponding practical applications have been further discussed and added in the revised Discussion section, quoted as follows:

“...Another key benefit of our framework is its exceptionally fast response time, performing OPC at the speed of light propagation, a feature paramount for numerous wavefront shaping (WFS) applications. An optimal WFS technique for scattering suppression would need to combine rapid system response, power efficient generation of the conjugated wave, and extensive degrees of freedom for precise wave manipulation^{15,61}. Speed is especially crucial for WFS, as it must be accomplished within the speckle correlation period, typically <1 ms, to be compatible with the dynamic nature of various specimens, such as living organisms and tissue samples. This requirement poses a challenge for conventional

*DOPC methods due to inherent delays in their operation. In contrast, the presented diffractive OPC processor, being entirely optical, achieves OPC operation as the light is transmitted through a thin optical volume, thus effortlessly satisfying the crucial demand for rapid response. By incorporating a diffraction efficiency penalty in the training loss function and enhancing the system's degrees of freedom through an increased number of diffractive features, our method holds the potential of achieving even higher power efficiency (see e.g., **Figs. 8-9**), also broadening the scope of operation beyond the limits of traditional OPC methods. These advantages render our framework particularly well suited for applications in biomedical imaging and potentially for astronomical observations through scattering media such as the atmosphere, where energy efficiency is critically important⁶²...*

6. About the deep learning part: in the training loss, MAE and MSE are used to guide the convergence of network for phase and amplitude, respectively, which is believed to be the proper choices to refine the modulation in the authors' practice. Nevertheless, the presented results still show inefficient amplitude modulation. Is it due to the insufficient training or is it one of the drawbacks in the terahertz realizations?

-- We will combine our answer here with the following question #7 as they are related to each other.

7. Following the last question, in the reflected realizations, the layers of networks are equivalently doubled but weaker amplitude modulation can be seen (Fig. 9), compared to the transmitted mode. So, does the accumulated attenuation or fabrication error among those diffractive layers limit or even weaken its performance?

-- We appreciate the reviewer's queries regarding the aspects of deep learning-based optimization and the comparative performance of the transmissive and reflective designs. Regarding the optimization between amplitude and phase in our models, we would like to clarify that this outcome did not result from insufficient training but from a deliberate trade-off to enhance the overall system performance. Moreover, the amplitude error can be mitigated by adjusting the hyperparameter that governs the amplitude uniformity penalty in our training loss function. A new analysis, which compared phase error values and amplitude error values between the reflective and transmissive designs using different hyperparameters, has been added to the manuscript, as shown in the newly added **Fig. 10b** and the new **Supplementary Fig. S8**. The contents related to this comparison, together with **Fig. 10b** and **Supplementary Fig. S8**, have been elaborated upon in the Discussion section of our revised manuscript, as quoted below:

*"...In our Results section, we successfully demonstrated the generation of phase-conjugated output fields using diffractive OPC processors. Nonetheless, some imperfections can be observed in the output amplitude profiles, especially in the reflective design of the diffractive phase-conjugate mirror. **These non-uniformity-related errors in the output amplitude profiles can be mitigated and suppressed by adjusting the hyperparameter that governs the amplitude uniformity penalty in our training loss function (see the Methods section for details).** To highlight this opportunity, we provide further analysis of this amplitude non-uniformity in **Fig. 10b** based on the transmissive and reflective designs presented in **Fig. 2b** and **Supplementary Fig. S7**, respectively, utilizing input fields composed of two randomly selected Zernike polynomials. By changing the penalty threshold (V) associated with the level of non-uniformity in the output amplitude field, we designed different OPC processors and plotted the corresponding changes in the average phase MAE values against the average amplitude MAE values. Notably, for the transmissive design, the amplitude error decreased*

from $8.89\% \pm 2.04\%$ to $4.51\% \pm 0.76\%$, alongside a relatively small phase error increase from $1.38 \pm 0.12\%$ to $1.97 \pm 0.15\%$. A similar pattern is also observed for the reflective OPC designs, with the amplitude error dropping from $9.41 \pm 2.23\%$ to $4.60 \pm 0.78\%$ and the phase error rising from $1.25 \pm 0.11\%$ to $1.67 \pm 0.13\%$. These results indicate that more uniform amplitude profiles can be achieved at the output aperture of our diffractive OPC processors with only a minor compromise in output phase accuracy. Moreover, our analyses suggest that reflective designs outperform their transmissive counterparts in terms of both amplitude and phase errors, which can be attributed to their double-pass configuration that more efficiently exploits the available degrees of freedom within the diffractive volume. These findings are further supported by the visualization examples provided in **Supplementary Fig. S8**, showing enhanced amplitude uniformity when the amplitude penalty threshold (V) is adjusted.”

Furthermore, we would like to emphasize that these observations discussed above do not stem from attenuation or manufacturing inaccuracies. The content related to the absorption was already discussed in the Methods section of our manuscript, quoted below:

“... For the diffractive OPC designs used for experimental validation, n and κ were set based on measurements using a terahertz spectroscopy system³⁹. In the other diffractive OPC designs used for numerical analyses at a single wavelength in the terahertz spectrum, the refractive index n was set as 1.7, while κ was chosen as 0. As for the diffractive multi-wavelength OPC designs shown in **Supplementary Fig. S1b**, the refractive index profile $n(\lambda)$ was set based on the dispersion of N-BK7 glass⁵⁴. $\kappa(\lambda)$ was set to 0, as the absorption of this material within the visible spectrum is negligible. ...”

To conclude, we sincerely thank the editor and reviewers for their constructive comments and feedback, which helped us to further improve the quality and clarity of our manuscript. We look forward to hearing back from you regarding our revised submission.

REVIEWER COMMENTS

Reviewer #1 (Remarks to the Author):

The manuscript was revised with very much improvement. A lot more analysis and new demonstrations especially with multiple wavelengths.

I agree that the manuscript is worth publishing but in a specialized journal. I would be happy to see it in Nature Communication if at least a real-world application of the technique is demonstrated. Current proof-of-concept demonstrations are also good but very much expected with machine learning approach.

Reviewer #2 (Remarks to the Author):

The authors have done an excellent job revising the manuscript and addressing all the queries appropriately. In my opinion, the manuscript is now suitable for publication.

Reviewer #3 (Remarks to the Author):

The reviewer appreciates this revision since more comprehensive demonstrations for diffractive OPC processor have been provided in this revision, including: 1) broadband/multi-wavelength OPC; 2) generalization validation with more compound optical patterns; 3) overcoming certain mechanical misalignments by a "vaccination" training strategy. It technically ensures that this framework can practically provide its unique benefits in realizing OPC.

Yet, one issue is left to be unaddressed as I commented in the last round. That is how fast the diffractive OPC can respond, which is a critical parameter to evaluate the performance of OPC. Although the authors have discussed the potential fast response of diffractive OPC processor, no clear evidence in the current version can support their claims.

We sincerely thank the referees for their reviews and the constructive feedback that we have received on our manuscript “**All-Optical Phase Conjugation Using Diffractive Wavefront Processing**” (NCOMMS-23-52142A)

As detailed below, we have revised our manuscript in response to the reviewer comments. The changes that we have made in the manuscript file have been highlighted in **yellow**.

We briefly summarize below the changes that we have made to our manuscript.

Summary of our Revisions:

- We have included an additional analysis to highlight the optical response time of our diffractive OPC processor designs under different wavelengths.

Revised sub-section:

- Discussion
-

Reviewer #1 (Remarks to the Author):

The manuscript was revised with very much improvement. A lot more analysis and new demonstrations especially with multiple wavelengths.

-- We sincerely thank the reviewer for his/her positive evaluations and constructive feedback provided during the review process.

I agree that the manuscript is worth publishing but in a specialized journal. I would be happy to see it in Nature Communication if at least a real-world application of the technique is demonstrated. Current proof-of-concept demonstrations are also good but very much expected with machine learning approach.

-- We would like to kindly reiterate the novelty and significant advantages of our diffractive OPC processor, which distinguish our work from existing literature. The key novelties and advantages include:

(1) In this work, we have experimentally demonstrated the precise control over the output complex field, which presents a significant challenge in diffractive optical information processing, approximating a nonlinear OPC function that has not been explored yet in the literature. To address this challenge, we created a new optimization process to minimize a specially devised loss function (detailed in Methods), which compares both the normalized amplitude and phase differences between the diffractive output fields \mathbf{o} and their corresponding ground truth $\mathbf{o}^{(GT)}$. Details regarding the training loss function, numerical forward model and structural parameters of the diffractive OPC processor are provided in the Methods section.

(2) Our study showcases **the first demonstration of a passive broadband OPC system** capable of phase conjugation across multiple distinct wavelengths, mitigating wavelength-specific limitations. This pivotal capability significantly differs from previous methodologies in OPC and creates a solid alternative to wavefront shaping and aberration compensation solutions across a large set of distinct wavelengths, where at each wavelength channel, the aberrated input wavefronts are independent

of the other wavelength channels, representing a highly challenging broadband OPC task. To substantiate the effectiveness of our approach, we have designed a multi-wavelength OPC system and conducted numerical simulations and experimentally validated our framework. See **Figure 7** as well as **Supplementary Figs. S1, S2, S3 and S4**, together with the subsection titled **“Optical phase conjugation of multi-wavelength illumination”**.

(3) The presented diffractive OPC platform holds design flexibility since it can function **at different parts of the electromagnetic spectrum** by simply scaling the design in proportion to the illumination wavelength without the need for retraining or fine-tuning. This feature makes our diffractive OPC framework highly desirable for correcting wavefront distortions at various parts of the spectrum where cost-effective and easy-to-implement phase conjugation solutions do not readily exist, such as the IR and THz bands. Moreover, since isotropic dielectric materials are used to fabricate these diffractive layers, the **OPC function is independent of the polarization state of the incident light**, which remains unchanged at the output of the diffractive OPC system. Combining these unique advantages, our diffractive OPC framework holds promise for various applications, including e.g. turbidity suppression and phase aberration correction, and can unlock different opportunities across diverse areas, including but not limited to biomedical imaging, microscopy, telescope systems and optical communication.

(4) Our diffractive OPC processor enables **wavefront reshaping at the speed of light, also high power efficiency (see Figures 8 and 9 of the main text)** — a capability not available in previous nonlinear crystal-based analog OPC systems. Some of these advantages of our diffractive OPC processors have been discussed in our Discussion section, as quoted below:

*“...Another key benefit of our framework is its exceptionally fast response time, performing OPC at the speed of light propagation, a feature paramount for numerous wavefront shaping (WFS) applications. An optimal WFS technique for scattering suppression would need to combine rapid system response, power efficient generation of the conjugated wave, and extensive degrees of freedom for precise wave manipulation^{15,61}. Speed is especially crucial for WFS, as it must be accomplished within the speckle correlation period, typically <1 ms, to be compatible with the dynamic nature of various specimens, such as living organisms and tissue samples. This requirement poses a challenge for conventional DOPC methods due to inherent delays in their operation. In contrast, the presented diffractive OPC processor, being entirely optical, achieves OPC operation as the light is transmitted through a thin optical volume, thus effortlessly satisfying the crucial demand for rapid response. To shed more light on this, we calculated the optical response time of our transmissive OPC processors following the same design illustrated in **Fig. 2b**. Here, the response time is determined by the light propagation time throughout the entire diffractive volume (from the input plane to the output plane), encompassing all the free space propagation between the layers as well as within the diffractive layer materials. Specifically, with an axial spacing of 12λ between successive layers and a base thickness of 0.2 mm for each diffractive layer, the OPC response times can be calculated as **~293 ps** and **~123 ps** at $\lambda = 0.75 \text{ mm}$ and $\lambda = 0.3 \text{ mm}$, respectively. For our diffractive phase conjugate mirror designs, which entail a double-pass configuration, the*

*optical response times will be doubled compared to their transmissive counterparts. For diffractive OPC designs operating at the visible and IR part of the spectrum, for example, the optical response times will be further reduced by more than one order of magnitude as the thickness of the diffractive processor volume can be significantly reduced due to the shorter wavelength. This significant speed advantage compared to existing AOPC and DOPC systems reported in the literature^{10,12,15} underscores the utility of our diffractive designs for applications that necessitate ultra-fast OPC response. Furthermore, by incorporating a diffraction efficiency penalty in the training loss function and enhancing the system's degrees of freedom through an increased number of diffractive features, our method holds the potential of achieving even higher power efficiency (see e.g., **Figs. 8-9**), also broadening the scope of operation beyond the limits of traditional OPC methods. These advantages render our framework particularly well suited for applications in biomedical imaging and potentially for astronomical observations through scattering media such as the atmosphere, where energy efficiency is critically important⁶².*

Finally, the novelty and the unique advantages of our diffractive multi-wavelength OPC approach have been elaborated on in the Discussion section, which is quoted below:

*"...The presented diffractive OPC processor designs offer significant advantages with their multi-wavelength operation, distinctly setting them apart from conventional OPC methods. While conventional AOPC systems are typically limited to narrowband operation due to their dependency on wavelength-specific nonlinear coefficients of materials, our diffractive OPC processors employ basic dielectric materials without the need for a specific dispersion relationship (see **Supplementary Fig. S2**). Our diffractive designs allow for multi-wavelength OPC operations, facilitating efficient performance over a wider spectral range, independent from the dispersion characteristics of the diffractive materials. For example, we showcased a broadband diffractive OPC processor designed using N-BK7, a widely used borosilicate glass known for its high damage threshold and superior optical quality, making it well-suited for a vast spectrum of wavelengths in the visible. In applications that demand a higher damage threshold, fused silica emerges as a viable alternative for the diffractive layer material, with UV to near-infrared band transmission and negligible thermal expansion⁶⁰..."*

Reviewer #2 (Remarks to the Author):

The authors have done an excellent job revising the manuscript and addressing all the queries appropriately. In my opinion, the manuscript is now suitable for publication.

-- We sincerely thank the reviewer for his/her positive evaluations and constructive feedback provided during the review process.

Reviewer #3 (Remarks to the Author):

The reviewer appreciates this revision since more comprehensive demonstrations for diffractive OPC processor have been provided in this revision, including: 1) broadband/multi-wavelength OPC; 2) generalization validation with more compound optical patterns; 3) overcoming certain mechanical misalignments by a "vaccination" training strategy. It technically ensures that this framework can practically provide its unique benefits in realizing OPC.

-- We sincerely thank the reviewer for his/her positive evaluations and constructive feedback provided during the review process.

Yet, one issue is left to be unaddressed as I commented in the last round. That is how fast the diffractive OPC can respond, which is a critical parameter to evaluate the performance of OPC. Although the authors have discussed the potential fast response of diffractive OPC processor, no clear evidence in the current version can support their claims.

-- To address the reviewer's question, we have included an additional analysis to highlight the optical response time of our diffractive OPC processor designs under different wavelengths. This has been added to the Discussion section of our revised manuscript, quoted as follows:

"...To shed more light on this, we calculated the optical response time of our transmissive OPC processors following the same design illustrated in Fig. 2b. Here, the response time is determined by the light propagation time throughout the entire diffractive volume (from the input plane to the output plane), encompassing all the free space propagation between the layers as well as within the diffractive layer materials. Specifically, with an axial spacing of 12λ between successive layers and a base thickness of 0.2 mm for each diffractive layer, the OPC response times can be calculated as ~ 293 ps and ~ 123 ps at $\lambda = 0.75$ mm and $\lambda = 0.3$ mm, respectively. For our diffractive phase conjugate mirror designs, which entail a double-pass configuration, the optical response times will be doubled compared to their transmissive counterparts. For diffractive OPC designs operating at the visible and IR part of the spectrum, for example, the optical response times will be further reduced by more than one order of magnitude as the thickness of the diffractive processor volume can be significantly reduced due to the shorter wavelength. This significant speed advantage compared to existing AOPC and DOPC systems reported in the literature^{10,12,15} underscores the utility of our diffractive designs for applications that necessitate ultra-fast OPC response..."

To conclude, we sincerely thank the editor and the reviewers for their constructive comments and feedback, which helped us to further improve the quality and clarity of our manuscript. We look forward to hearing back from you regarding our revised submission.

REVIEWERS' COMMENTS

Reviewer #3 (Remarks to the Author):

The authors have addressed my concerns and I have no more comments at this point. Thank you.